# Membrane Domain Localization and Interaction of the Prion-Family Proteins, Prion and Shadoo with Calnexin

**DOI:** 10.3390/membranes11120978

**Published:** 2021-12-13

**Authors:** Divya Teja Dondapati, Pradeep Reddy Cingaram, Ferhan Ayaydin, Antal Nyeste, Andor Kanyó, Ervin Welker, Elfrieda Fodor

**Affiliations:** 1Institute of Biochemistry, Biological Research Centre, 6726 Szeged, Hungary; divya.medbio@gmail.com (D.T.D.); pkc23@med.miami.edu (P.R.C.); 2Doctoral School of Multidisciplinary Medical Sciences, University of Szeged, 6720 Szeged, Hungary; 3Hungarian Centre of Excellence for Molecular Medicine (HCEMM) Nonprofit Ltd., 6723 Szeged, Hungary; Ferhan.Ayaydin@hcemm.eu; 4Cellular Imaging Laboratory, Biological Research Centre, 6726 Szeged, Hungary; 5Proteoscientia Ltd., 3066 Cserhátszentiván, Hungary; nyeste.antal@ttk.hu; 6Institute of Enzymology, Research Centre for Natural Sciences, 1117 Budapest, Hungary; 7Biospirál-2006. Ltd., 6726 Szeged, Hungary; akanyo20@gmail.com

**Keywords:** lipid rafts, GPI-anchored proteins, endoplasmic reticulum, detergent-free raft isolation

## Abstract

The cellular prion protein (PrP^C^) is renowned for its infectious conformational isoform PrP^Sc^, capable of templating subsequent conversions of healthy PrP^C^s and thus triggering the group of incurable diseases known as transmissible spongiform encephalopathies. Besides this mechanism not being fully uncovered, the protein’s physiological role is also elusive. PrP^C^ and its newest, less understood paralog Shadoo are glycosylphosphatidylinositol-anchored proteins highly expressed in the central nervous system. While they share some attributes and neuroprotective actions, opposing roles have also been reported for the two; however, the amount of data about their exact functions is lacking. Protein–protein interactions and membrane microdomain localizations are key determinants of protein function. Accurate identification of these functions for a membrane protein, however, can become biased due to interactions occurring during sample processing. To avoid such artifacts, we apply a non-detergent-based membrane-fractionation approach to study the prion protein and Shadoo. We show that the two proteins occupy similarly raft and non-raft membrane fractions when expressed in N2a cells and that both proteins pull down the chaperone calnexin in both rafts and non-rafts. These indicate their possible binding to calnexin in both types of membrane domains, which might be a necessary requisite to aid the inherently unstable native conformation during their lifetime.

## 1. Introduction

The cellular prion protein (PrP^C^) is renowned for its conformationally aberrant isoform PrP^Sc^, which has infective traits exerted through a “self-replicating” ability, inducing and templating similar conversion of nearby healthy PrP^C^ molecules [1,2]. This process leads to amyloidal prion aggregates; cell death; and the group of incurable neurodegenerative diseases known as transmissible spongiform encephalopathies (TSEs) or prionopathies, which affect a wide range of mammals [3,4,5]. The most commonly known animal TSEs are scrapie of sheep and goat [6], bovine spongiform encephalopathy (BSA) [7] and chronic wasting disease (CWD) of deer and elk [8]. About 15% of human TSEs are linked to various mutations in the *PRNP* gene encoding PrP^C^ [9], leading to Creutzfieldt–Jacob disease, (CJD), Gerstmann–Straussler–Scheinker disease (GSS) or fatal familial insomnia (FFI), whereas the majority are sporadic (sporadic CJD and sporadic FFI) and a few are transmitted either iatrogenically (iCJD) or through the consumption of infected tissue (kuru, new variant CJD (nvCJD)) [10]. Common to TSEs is a rapid progression after detection and a convergence into fatal neurodegeneration, a process still unresolved despite many studies [11], leaving us with a lack of reliable early markers and an inability to cure TSEs.

Not less puzzling is the function of healthy PrP^C^s, for which no univocal cellular role is inferred. With PrP^C^ being expressed in many tissues, with the highest expressions in the central nervous system (CNS), lymphoid tissues and heart [12,13], and being highly conserved among species [14], its deletion neither is lethal nor produces obvious phenotypes in mice [15,16], cattle [17] or goat [18,19]. Its presence, however, is required for the acquisition of prion disease, as *Prnp*-KO mice are resistant to propagating infection and developing disease after intracerebral inoculation by infectious prions [20]. Furthermore, a multitude of binding partners had been reported for PrP^C^, including protein and non-protein interactors [21,22], rendering it a multifaceted and multitasking protein involved in several cellular processes: stress protection, metal ion homeostasis, cell differentiation, adhesion, neuronal growth, myelin maintenance, mitochondrial homeostasis, circadian rhythm and immune modulation [23,24]. Importantly, contrary to PrP^Sc^, PrP^C^ is mostly invoked in neuroprotective roles, exerted either directly or indirectly, participating via binding partners in cell signaling processes. Intriguingly, its protective roles in other amyloidal neurodegenerative diseases are emerging, not only as an antioxidant but also as a receptor for oligomers of the β-amyloid peptide and tau playing a major role in the pathogenesis of Alzheimer’s disease (AD) or binding some forms of α-synuclein, which are involved in the Lewy bodies of Parkinson’s diseases [25].

Nascent PrP^C^ possesses a signal sequence that targets it to the secretory pathway where it matures to a 208 residues-long glycoprotein (aa. 23–230, mouse numbering) with a glycosylphosphatidylinositol (GPI) anchor attachment and one or two complex N-glycosylations. At the outer leaflet of the cellular plasma membrane (PM), PrP^C^ is believed to reside mostly in lipid rafts, in the membrane domains known as dynamic specialized signaling platforms [26], as commonly observed for the GPI-anchored proteins [27]. With several transmembrane and extracellular interacting partners, it is regarded as a cell surface scaffold protein and a key player in several signaling processes linked to rafts [28]. The protein possesses a peculiar structure of two halves with opposing dynamics and folds: an unstructured N-terminal and a globular C-terminal domain [29,30], which overall confer plasticity and multistate stability to PrP^C^, a common attribute of amylogenic proteins [31,32]. Several characteristic and functionally important regions reside along its sequence (Figure 1). Its globular domain (aa. 126–230) has a highly conserved, mostly helical tertiary structure, with one intramolecular disulfide bond [33,34]. The unstructured N-terminal half possesses several regions involved in specific functions [35]. Among them, the highly conserved hydrophobic domain (HD, aa. 111–133) is the connecting segment to the globular part and serves as a key site where several of the prion protein’s interactions with partner proteins had been mapped [22]. The octapeptide repeat region (OR, aa. 59–90) confers to PrP the ability to bind Cu^2+^ and Zn^2+^ [36,37,38]. The basic N-terminal patch (aa. 23–26) together with OR exert regulatory effects upon transmembrane proteins [39,40] and establish a regulatory cis-interaction with the globular domain [41]. The polybasic N-terminal region (aa. 23–31) and a second positively charged patch (aa.100–109) were also reported to bind nucleic acids, which is viewed as maybe contributing to TSEs [42,43].

Two other genes were discovered to belong to the prion family proteins, *PRND* and *SPRN*, encoding doppel (Dpl) [44] and Shadoo (Sho) [45] proteins. The functional relations, if any, between prion protein (PrP) and these paralogs is also not well understood. Structurally, each resembles one of the halves of PrP: Dpl being globular and Sho disordered in terms of folds, while both are GPI-anchored and complex glycosylated as PrP [31,46]. Contrary to Dpl, Sho is also primarily expressed in the CNS as PrP^C^, from the early embryonic stage to the adult state [47,48,49]. It was shown to play a role in embryonic development and tissue formation [50,51] and to likely participate in overlapping embryonic pathways with PrP^C^, as *Sprn* mRNA knockdown in *Prnp*^0/0^ mice proved lethal [48]. However, its absence in *Sprn*^0/0^ or in double-knockout (*Sprn*^0/0^*-Prnp*^0/0^) mice proved otherwise, resulting in no dramatic phenotypes [52,53] and rendering Sho’s cellular role also puzzling. Many candidate interacting partners had been proposed for Sho as well [21], among them several common with PrP; moreover, they are confirmed binding partners themselves [54,55]. All of these hint at a less univocal cellular role for Sho, perhaps similarly to PrP; however, further studies and/or confirmation of more of its proposed binding partners and in different systems is needed in order to clarify this.

Albeit with low sequence homology, Sho similarly harbors a highly conserved hydrophobic domain (Figure 1), which is its only homologous sequence to PrP—to the PrP’s conserved HD, responsible for many important protein–protein interactions of PrP, including homodimerization [56]—and in which Sho has also been proven to be the site for interaction with PrP [54]. At the N-terminal to its HD, Sho possesses an endo-proteolytic cleavage site resembling the α-cleavage site of PrP. However, Sho lacks a copper-binding region, indicative of a potential for distinct functions from PrP^C^, harboring instead arginine-rich tetra-repeat segments towards its N-terminal, conferring the ability to bind RNA and nucleic acids [57,58]. Additionally, this region plays a role in the nuclear localization of Sho as we demonstrated earlier [59] in different cells expressing the protein, and, in line with this, nuclear localization of Sho as a response to proteasome inhibition was recently reported, invoking the role of the arginine-rich repeat region in this process [60].

Sho is inferred to have similar neuroprotective actions as the wild-type PrP^C^ in several experimental settings. In cell systems, Sho is found to rescue cells against the toxic effects of Dpl or of Shmerling deletion mutant Δ[32–121]PrP in primary cerebellar granule neuronal (CGN) cultures from *Prnp*^0/0^ mouse [47,61] and to protect cells against the excitotoxic stress exerted by glutamate and the toxic effects of the hydrophobic domain deletion mutant, Δ[113–133], PrPΔHD in human SH-SY5Y cells [62]. In addition, studies by our group demonstrated that, contrary to wild type PrP^C^, Sho expression sensitizes human SH-SY5Y and HEK293 cells to certain drugs, similar to the toxic, central region (CR)-deletion mutant, Δ[105–125], PrPΔCR—a phenotype that could be rescued by the wild type PrP^C^ [63]. Additionally, we showed that the expression of Sho produces spontaneous large inward currents in cell cultures, a similar effect that is exerted by the HD-deletion mutant PrPΔHD [64], which could also be rescued by PrP^C^ in both cases. In animal models, Sho and PrP are observed to be involved in embryonic development; however, a cross regulation between the levels of the two proteins is not apparent in the adult state [47]. Additionally, although Sho knockdown experiments on the PrP-null background were lethal, implying overlapping functions for the proteins [48,51], double-knockout experiments could not prove that the two proteins are functional homologs [53]. In disease conditions, decreased levels of Sho in the brain were reported in several experimental prion-infected rodent models and sheep [47,65,66], echoing the depletion of PrP^C^ during prion disease progression [67]. However, Sho neither proved to play a protective effect against infection nor was essential for TSE development, and its overexpression did not influence prion replication kinetics in transgenic mice [65,68,69]. Additionally, contrary to PrP^C^, no reduction in Sho was observed in transgenic models of other neurodegenerative diseases as of Alzheimer’s and Parkinson’s diseases or frontal dementia, expressing the disease-related mutant β-amyloid precursor protein, α-synuclein or tau [47,65].

Raft localization at the plasma membrane is known to serve GPI-anchored proteins in fulfilling their specific functions in signaling. This had been confirmed in numerous cases for PrP^C^ [28]: it was shown to bind within rafts to the laminin precursor protein and the laminin receptor and participates in neuritogenesis [70], recruits the neuronal cell adhesion molecule to rafts to promote neurite outgrowth and cell survival [71], and binds to STI1 to engage in downstream neuroprotective activities [72,73]. Sho has also been proposed to act as a cell surface receptor for hyaluronate and/or extracellular RNA and engages in signaling processes [57]. On the other hand, raft localization plays an important role in prion infection, although the exact mechanism and location of the conversion of PrP^C^ by PrP^Sc^ are not clear [26,74]. For both PrP^C^ and Sho, raft localization proved to be crucial for their correct folding as well. The disruption of rafts by cholesterol depletion resulted in an impairment of the proper folding of PrP^C^ with the accumulation of a partially proteinase-K (PK) resistant misfolded form in the early secretory pathway [75]. Similarly, for Sho, an accumulation of a PK resistant form and an increase of the unglycosylated form in the endoplasmic reticulum (ER) had been reported upon raft-disruption [76]. Interestingly, the same study found that a percentage of Sho, contrary to PrP, is in a partially PK-resistant, aggregated state already at natural conditions of the cells when using primary neuronal GT1 and human neuroblastoma SH-SY5Y cells. This percentage of Sho increased upon disrupting rafts by cholesterol depletion (where similar forms of PrP also appear). Moreover, the authors also demonstrated that, in parallel to partial PK-resistance, both the mature and the unglycosylated forms of Sho co-immunoprecipitate (co-IP) with the ER-chaperone calreticulin (CRT), a binding that was enhanced upon disrupting rafts by cholesterol depletion [76]. Furthermore, earlier studies on PrP found that another ER-chaperone, calnexin (CNX), could bind full-length PrP both in vitro, preventing its thermal aggregation, and in vivo, as it Co-IPed with PrP^C^. In vivo, the proteins co-immunoprecipitated either when endogenous or when co-transfected from cell lysates using 293T and human neuroblastoma SK-N-SH cells. In the latter case, this binding to CNX was shown to prevent the cytotoxicity of PrP in these cells [77]. While intriguing in itself that ER chaperones attach to mature GPI-anchored proteins, it remains unknown, however, whether such an interaction of PrP and CNX is preferred in specific membrane domains versus others. It is also not known if Sho displays binding to CNX, similar to PrP, and in which membrane domains such an interaction would occur.

Here, we set forward to study the raft-localization and membrane-domain distribution of the two prion family proteins, the prion protein and Shadoo, and their possible interaction with calnexin, applying a non-detergent-based raft isolation method and transgenic N2a cells expressing the proteins. Our results reveal that Sho and PrP occupy both raft- and non-raft-type membrane domains, with essentially similar distribution patterns along the gradient fractions. We found that the two prion family proteins pull down calnexin in both raft- and non-raft type membrane fractions, indicating that at least a fraction of these proteins maintains binding with calnexin while partitioning to different membrane domains during their normal cellular biology.

## 2. Materials and Methods

### 2.1. Materials

Unless otherwise specified, all chemicals and reagents, the ANTI-FLAG M2 affinity gel, the Immobilon-P PVDF Membrane (pore size: 0.45 μm) and Immobilon Western Chemiluminescent HRP Substrate were from Merck/Sigma-Aldrich and Millipore (Darmstadt, Germany). All cell culture media and supplements, the TurboFect transfection reagent, the Amplex^®^ Red Cholesterol assay kit, the CellLight™ Golgi-RFP, BacMam 2.0 Golgi-labelling reagent, and the eight-well microscopy plates Nunc™ Lab-Tek™ II Chambered Coverglass were from Thermo Fisher Scientific (Waltham, MA, USA). Paraformaldehyde was from Riedel-de Haën (Seelze, Germany), the RC-DC Protein Assay kit from BioRad (Hercules, CA, USA), the ProSieve^®^ QuadColor™ protein marker from Lonza (Basel, Switzerland) and the PNGase F was from New England Biolabs (Ipswich, MA, USA).

### 2.2. Antibodies

The monoclonal anti-prion protein antibody SAF-32 (Cat. No. A03202) was purchased from Cayman Chemical (Ann Arbor, MI, USA). The anti-Shadoo polyclonal SPRN antibody (C-terminal) (Cat. No. AP4754b) was from Abgent (San Diego, CA, USA). The Living Colors EGFP monoclonal antibody (Cat. No. 632569) was from Takara Bio/Clontech Laboratories (Mountain View, CA, USA). The polyclonal anti-Calnexin antibody (Cat. No. ab10286) and anti-Nuclear Pore Complex polyclonal antibody (NPC) (Cat. No. ab73291) were purchased from Abcam (Waltham, MA, USA). Flotillin-1 (Cat. No. 610820) was from Fischer Scientific BD Biosciences (San Jose, CA, USA). The anti-Transferrin receptor antibody (TfRC) (Cat. No. SAB4200398); monoclonal ANTI FLAG M2-Peroxidase (HRP) Clone M2 antibody (Cat. No. A8592); and the secondary antibodies anti-Rabbit IgG (whole molecule)-Peroxidase antibody (A9169) and anti-Mouse IgG (Fab specific)-Peroxidase antibody (Cat. No. A3682), produced each in goat, were from Merck/Sigma-Aldrich (Darmstadt, Germany). The polyclonal goat anti-rabbit IgG (H + L) secondary antibody Alexa Fluor 568 conjugate Cat. No. A11011) and polyclonal goat anti-mouse IgG (H + L) secondary antibody Alexa Fluor 488 conjugate (Cat. No. A10667) were from Thermo Fisher Scientific (Waltham, MA, USA).

### 2.3. Plasmid Constructs

The plasmid, pCMV3-C-OFPSpark, encoding for mouse calnexin with a C-terminal OFPSpark-tag and on a CMV promoter, used for the generation of cells transiently expressing red fluorescent protein-tagged calnexin, was purchased from Sino Biological (Beijing, China) (Cat. No. MG53126-ACR). To produce the Shadoo expressing N2a Sho-EYFP and its control EYFP cells, the same plasmids were used as in our earlier work [59], in which the enhanced yellow fluorescent protein (EYFP) coding DNA sequence (CDS) was used as a fusion tag to Sho, inserted between the protein’s (Sho) C-terminus and its GPI-signal peptide coding sequence or, for the control protein and control cells, between the ER-targeting signal sequence and the GPI-signal sequence of Sho, respectively. These plasmids were also used as starting constructs to generate two additional plasmids: p_mSho-EYFP-FLAG-GPI(mSho) (Appendix A) and p_SS(mSho)-EYFP-FLAG-GPI(mSho) (Appendix A), where a FLAG tag followed by two Strep-Tag II-coding sequences were also inserted between the EYFP and the GPI-signal sequence of Sho in both the Sho and the control protein expression plasmids. These plasmids were then used to generate the stable N2a cells named Sho-EYFP-FLAG and EYFP-FLAG cells, respectively, utilized in co-immunoprecipitation experiments. Since in the final experiments only the FLAG sequence was exploited for this purpose, from here onwards, we omit indicating the presence of Strep-Tag II from the notations and naming of the cells and samples. For the generation of the stable transgenic N2a PrP-EGFP and its control EGFP transgenic cells, expressing EGFP-tagged mouse PrP and its corresponding control protein without PrP, respectively, plasmids were constructed similar to that for Shadoo, as follows: for mPrP expression, the p_mPrP-EGFP-GPI(mPrP) plasmid was generated utilizing an EGFP cassette as a fusion tag, inserted in between the C-terminal end and the GPI-signal coding sequence of mPrP (Appendix A). For the expression of the control protein, the EGFP cassette was inserted to be flanked by the ER-targeting and the GPI-signal sequences of PrP in a similar vector backbone as for mPrP (Appendix A). To generate the stable N2a cells expressing untagged PrP and soluble EGFP (simultaneously, but not in fusion), named N2a/PrP(+EGFP) cells, and its corresponding control cells, N2a(EGFP), the plasmids constructed and reported earlier [63] were used.

### 2.4. Cell Culturing

Neuro-2a (N2a) mouse neuroblastoma cells were purchased from ATCC (CCL-131TM) (Manassas, Virginia, USA). Cells were cultured typically in Dulbecco’s modified Eagle medium with high glucose (4.5 g/L) (DMEM) supplemented with 10% heat-inactivated fetal bovine serum (FBS), 100 units/mL Penicillin and 100 µg/mL Streptomycin and 1% GlutaMAX, at 37 °C in a humidified atmosphere with 5% CO_2_. The cells were passaged at 90–95% confluence at a 1:20 splitting ratio.

### 2.5. Generation of Stable Transgenic Cells

#### 2.5.1. N2a PrP(+EGFP) and N2a(EGFP) Cells

Stable transgenic N2a cells expressing untagged mouse PrP protein and soluble EGFP (named N2a/PrP(+EGFP) cells) and its control cells (named N2a(EGFP)) expressing soluble EGFP alone were established using the Sleeping Beauty gene delivery system, as we described earlier [63]. Briefly, 48 h prior to transfection, 5 × 10^4^ N2a cells/well were seeded on a six-well plate. The transfection was carried out at 50–70% confluence, using 3.5 µg of circular plasmid DNA with the TurboFect transfection reagent, in accordance with the manufacturer’s protocol. The EGFP and PrP, and EGFP encoding pSB transposon vectors used for transfections are described in Nyeste et al. (2016) [63]. An amount of 3 μg Transgene encoding pSB vectors was used in each transfection condition, as well as 0.5 µg of Transposon encoding SBx100 or inactive transposon encoding SB6 vector. Fluorescence-activated cell sorting (FACS) was used to separate the cells with stable transgene expression in the transfected cell populations at day 14 post-transfection. To avoid the positional effect of the integrated transgenes instead of establishing single-cell clones, a stable population of transgenic cells was propagated. Parental and transgenic N2a cell lines were regularly tested for mycoplasma contamination, and EGFP expression was examined at every passage. The experiments were carried out on cultures in which at least 90% of the cells expressed EGFP.

#### 2.5.2. N2a PrP-EGFP, Sho-EYFP, Sho-FLAG-EYFP and Their Respective Control Cells (EGFP, EYFP and EYFP-FLAG) Cells

Stable transgenic N2a cells were established by transfection of cells using the respective DNA plasmid via the TurboFect transfection reagent, according to the manufacturer’s protocol. Briefly, parental N2a cells were plated a day before transfection on eight-well chambered cover glass-bottomed plates, in 250 µL culture media per well, typically as 3–3.5 × 10^4^ cells/well, to reach 50–60% confluence after 24 h. Transfection was carried out using 0.25 μg of plasmid DNA and 0.5 μL of TurboFect per well of cells. After 6 h of culturing, the media were replaced by fresh growth media and the cells were cultured for an additional 12–18 h. Next, the cells were treated with 500 μM of geneticin and were cultured for two more days prior to transferring them to 6 cm cell culture Petri dishes. The cells were grown under geneticin selection for at least 10 days, while the media with antibiotic were replaced with fresh media every second day. Usually, after the first week, cells were spread into 100 mm diameter Petri culture dishes to form well-separated individual colonies. Individual EGFP-positive colonies were identified, picked under a fluorescence microscope in a sterile hood. Eight colonies of Sho-EYFP cells and 10 colonies of PrP-EGFP cells were picked and transferred to individual wells of a 48-well plate. The colonies were cultured further without applying any antibiotic selection and were grown until they reached confluence in 100 mm Petri dishes. The cell populations were tested under a fluorescence microscope, as well as analyzed and sorted for EGFP-positive cells using BD FACSJazz fluorescence-activated cell sorter instrument (BD, Franklin Lakes, NJ, USA). Sorting generally occurred not earlier than 3 weeks after transfection. Individually sorted colonies were maintained separately as frozen stocks until used. Mixed populations were generated by mixing the individual colonies in equal ratios: five colonies in the case of Sho-EYFP cells, 10 colonies in the case of PrP-EGFP cells, seven colonies in the case Sho-EYFP-FLAG cells and 13 colonies in the case of EYFP-FLAG cells to rule out eventual clonal bias in the outcome of the experiments.

### 2.6. Transient Transfection of Cells

For transient overexpression of calnexin, the cells were seeded on eight-well coverslip glass-bottom plates at 3.5 × 10^4^ cells/well density in 250 µL complete DMEM, one day prior to transfection. The cells were transfected using 0.38 µg of plasmid DNA and 0.6 µL of TurboFect per well, first mixing the DNA and transfection reagent in serum-free DMEM, incubating the mixture at room temperature (RT) for 20 min, then gently pipetting, and evenly spreading it on top of the cells. Cells were imaged usually between 48 to 72 h after transfection.

### 2.7. Golgi Complex Labelling and Confocal Microscopy

Transgenic N2a Sho-EYFP or PrP-EGFP cells along with parental N2a cells were labeled for Golgi complex using CellLight™ Golgi-RFP, BacMam 2.0 reagent according to the manufacturer’s instructions. Briefly, 0.4 × 10^5^ cells were plated on eight-well chambered cover glass-bottomed plates a day before the addition of the reagent to the cells. On the next day, the reagent (of 1 × 10^8^ particles/mL) was added to the cells by mixing 12 μL of the reagent with the growth media to give a PPC (particles per cell) value of 30 for the final concentration of the reagent, as suggested by the protocol, and the cells were further cultured overnight in a CO_2_ incubator at 37 °C. Images of the labeled cells were acquired using a Fluoview FV1000 (Olympus Life Science Europa GmbH, Hamburg, Germany) confocal laser scanning microscope using 405 nm, 488 nm and 543 nm lasers for excitation of DAPI, EGFP/EYFP and RFP with emission filters of 425–475 nm, 500–530 nm and LP560, respectively. The images were taken using an UPLSAPO 20x (N.A. 0.75) objective, applying 4.0 μs/pixel sampling speed and sequential, unidirectional scanning mode.

### 2.8. Extraction of Total Cell Lysates

Cells grown to 80% confluence in one 100 mm Petri plate for each type of cell were washed with ice-cold phosphate-buffered saline (PBS) (137 mM NaCl, 2.7 mM KCl, 6 mM Na_2_HPO_4_.2H_2_O, 1.4 mM KH_2_PO_4_ and pH 7.4), were scraped in 5 mL of PBS and were pelleted at 500× *g* for 5 min. The supernatant was discarded, and 1 mL of cold lysis buffer (50 mM Tris–HCl pH 7.4, 150 mM NaCl, 1 mM EDTA, 1% Triton X-100, 1 mM phenylmethyl sulfonyl fluoride and protease inhibitor cocktail) was added to the pellet. The resuspended pellet was transferred to 1.5 mL microfuge tubes and was kept on a rocker for 30 min at 4 °C to extract the proteins. The samples were then centrifuged at 20,000× *g* for 10 min at 4 °C, and the supernatant was collected as the total cell lysate.

### 2.9. Detergent-Free Separation of Lipid Rafts

Before proceeding to isolate lipid rafts, the fluorescent protein expression of cells was examined by microscopy and FACS analysis. The experiments were carried out on cultures where above 90% of the population expressed the fluorescent proteins. Cells were seeded at 1 × 10^6^ cells per 100 mm diameter Petri dish, with each cell type in at least 10 to 12 plates. After 24 h, the cells were harvested for membrane raft separation using the detergent-free OptiPrep-density gradient method of Macdonald and Pike [78], briefly, as follows. All procedures were carried out on ice. For each type of cell, 10 uniformly grown plates of cells were washed twice with ice-cold PBS, and the cells were scraped into 2 mL of Buffer A (20 mM Tris-HCl, pH 7.8, 250 mM sucrose, 1 mM CaCl_2_ and 1 mM MgCl_2_) and were pelleted by centrifugation at 250× *g* for 2 min. The cell pellets were resuspended in 1 mL of Buffer A containing protease inhibitors at final concentrations as follows: 0.2 mM aminoethylbenzenesulfonyl fluoride, 1 µg/mL aprotinin, 10 µM bestatin, 3 µM E-64, 10 µg/mL leupeptin, 2 µM pepstatin and 50 µg/mL calpain inhibitor I. The cells were then lysed by passage through an 18 G × 1.5ʺ needle 30 times, for each sample. The lysates were centrifuged and the post-nuclear supernatants were collected and transferred to new tubes. The pellets were again lysed and centrifuged similarly as before. The resulting second post-nuclear supernatant was mixed with the first. The total protein concentration of the combined sample was determined by a DC protein assay kit (BioRad). Samples of 5 mg total protein content (generally for all cell types) were used for separation, which was mixed with a Base buffer composed of 50% OptiPrep in 20 mM Tris-HCl, pH 7.8 and 250 mM sucrose, to give a final concentration of 25% OptiPrep and a final volume of 4 mL and was placed at the bottom of a 12 mL ultracentrifuge tube. Then, 8 mL of a continuous gradient of 0–20% OptiPrep in Base buffer was layered on top of the 25% OptiPrep-sample solutions in the ultracentrifuge tubes. The gradients prepared were ultra-centrifuged for 90 min at 52,000× *g* using a TH641 rotor in a Sorvall ultracentrifuge (Sorvall WX 80 + Ultracentrifuge, Thermo Fisher Scientific, Waltham, MA, USA) at 4 °C. In total, 18 fractions of 0.67 mL were collected starting from the top of the gradient from each tube. Equal volumes from each fraction were subjected to sodium dodecyl sulfate-polyacrylamide gel electrophoresis (SDS-PAGE) and Western blot analysis of the selected proteins. The total cholesterol in each fraction was determined using the Amplex^®^ Red Cholesterol assay kit. A blank density gradient (where the volume of the cell lysate was replaced with Buffer A) was run in parallel to determine the density of the OptiPrep gradient corresponding to each fraction, where the density of each gradient fraction was determined by measuring the absorbance of OptiPrep at 340 nm by a Nanodrop-1000 spectrophotometer. Raft- and non-raft-type membrane fractions were identified based on Persaud-Sawin et al. (2009) using Western blotting for proteins, such as flotillin-1 as a resident protein known as to be enriched in rafts and transferrin receptor protein (TfRC) as a non-raft resident protein, as well as monitoring the total protein content and total cholesterol content of the fractions. Rafts were considered those with high cholesterol and low total protein content and for which TfRC was absent and Flottilin-1 was present.

### 2.10. PNGase F Treatment

PNGase F treatment was performed on total post-nuclear membranes of cells, prepared as described under 2.8 detergent-free separation of lipid rafts. Samples of total cell lysate were subjected to deglycosylation of the total proteins with PNGase F (Peptide -*N*-Glycosidase F) enzyme according to the manufacturer’s protocol. Briefly, two parallel aliquots of 20 μg of the total protein amount from each sample were denatured at 100 °C for 10 min, one of the aliquots was treated with 1500 units of PNGase F enzyme, and the other was left untreated as a control. Both samples were incubated at 37 °C for 2 h. The samples were subjected to SDS-PAGE using 12 or 8% PA SDS gels, and deglycosylation of the proteins was assessed by Western blotting.

### 2.11. Western Blotting

Western blotting was performed from either total cell lysate (typically, 4 µg of the total proteins/sample) or isolated membrane fractions. In the case of the membrane fractions, from each gradient fraction obtained after lipid-raft isolation, an aliquot of 10 µL was used for the Western blot analyses of selected proteins. The 10 μL samples were denatured by 2.5 μL of 5× Laemmli sample-buffer (250 mM Tris-HCl pH 6.8, 10% SDS, 0.02% bromophenol blue, 30% glycerol and 5% β-mercaptoethanol) at 100 °C for 5 min and were loaded on to 8%, 10% or 12% polyacrylamide (PA)-SDS gels depending on the size of the protein of interest. Gradient samples from fraction numbers 1 through 18 were loaded on two separate gels with the same percentages in parallel (fraction numbers 1 through 12 on one gel and fraction numbers 13 through 18 on the other gel) and were subjected to SDS-PAGE for 60 min at 150 V. The separated proteins on the two gels were electro-blotted onto single methanol-activated Immobilon^®^–P PVDF transfer membrane, one beside the other, such that samples from fraction numbers 1 to 18 were in line on the blot. The electro-blotting was conducted for 1 h at constant current (400 mA) in cold transfer buffer (25 mM Tris, 192 mM glycine, and 20% methanol, pH 7.4). The membrane was then blocked with 5% non-fat milk in PBS with 0.05% Tween-20 (Blocking buffer) for 1 h at RT, washed three times with PBS with 0.05% Tween-20 (PBST) for 5 min, and incubated with the corresponding primary antibodies overnight at 4 °C. The following primary antibodies and dilutions were used in the experiments: anti-prion protein, SAF-32 (1:3000); anti-Shadoo *SPRN* (1:4000); anti-GFP (1:4000); anti-flotillin-1 (1:4000); anti-transferrin receptor, TfRC (1:4000); anti-calnexin (1:4000) and anti-nuclear pore complex protein, NPC (1:4000). After primary antibody incubation and washing with PBST four times (5 min each wash) to remove the unbound antibodies, the membrane was incubated with the corresponding horseradish peroxidase-conjugated secondary antibodies in the Blocking buffer for 2 h at RT, typically at 1:60,000 dilution. The unbound secondary antibodies were removed by washing the membrane for 5 min five times. Protein bands were detected by chemiluminescent HRP substrate and were visualized on the X-ray films.

### 2.12. Cholesterol Determination

The total cholesterol in each fraction was determined using the Amplex^®^ Red Cholesterol assay kit as per manufacturer instructions. Briefly, 50 μL of cholesterol reference standards, positive controls (hydrogen peroxide), negative control (only buffer) and even-numbered gradient fractions (fraction number: 2, 6, 8, 10, 12, 14 and 16) were placed in a 96-well flat-bottomed plate. Each sample was then mixed with 50 μL of Amplex Red reagent/HRP/cholesterol oxidase/cholesterol esterase working solution (300 μM Amplex Red reagent, 2 U/mL HRP and cholesterol oxidase, and 0.2 U/mL cholesterol esterase) and the plate was incubated for 30 min at 37 °C protected from light. Fluorescence was measured after 30 min using 565 nm excitation and 580 nm emission wavelengths in a Fluoroskan Ascent FL Microplate Fluorometer and Luminometer (Thermo Scientific brand, Thermo Fisher Scientific, Waltham, MA, USA)) microplate reader.

### 2.13. Immunocytochemistry

Parental N2a and transgenic PrP-EGFP, Sho-EYFP-FLAG and their control EGFP and EYFP-FLAG cells, respectively, were seeded on eight-well cover glass-bottom plates (Nunc, Lab-Tek II). After ~24 h, once the cell confluence reached 80–90%, the cells were washed twice with PBS and were fixed with 4% paraformaldehyde (PFA) for 7 min at RT. After fixing, PFA was removed and cells were washed three times by PBS before permeabilization by 0.1% Triton X-100 diluted in PBS for 7 min at RT. After permeabilization, Triton X-100 was removed from the cells by washing with PBS three times, after which the cells were blocked by adding blocking solution (1% bovine serum albumin (BSA) in PBS) for 1 h at RT. The cells were then incubated with primary antibodies against the prion protein (monoclonal SAF-32 for N2a, α-GFP for PrP-EGFP cells), Shadoo (α-GFP for Sho-EYFP-FLAG cells) and calnexin (polyclonal anticalnexin antibody, α-CNX) at 1:200 dilutions in blocking solution at 4 °C, overnight. On the next day, primary antibodies were washed from the cells three times using blocking solution, and the corresponding Alexa Fluor-conjugated secondary antibodies were applied to the cells as follows: for the prion protein and Shadoo, the anti-mouse Alexa Fluor 488-labelled (green) and, for calnexin, the anti-rabbit Alexa Fluor 568-labeled (red) secondary antibodies were applied each at 1:300 dilution in blocking solution for 1 h at 37 °C. The unbound secondary antibodies were washed out three times by PBS, and the nuclei were stained using 100 ng/mL of 4′, 6-diamidino-2 phenylindole HCl (DAPI) for 5 min at 37 °C. DAPI was washed from the cells by PBS, and images of the cells were taken in PBS. Immunofluorescent signals were acquired with a VisiScope CSU-W1 spinning disk confocal microscope (Visitron Systems GmbH, Puchheim, Germany) using 100× oil immersion objective and excitation lasers of 405 nm for DAPI, 488 nm for Alexa Fluor 488-labeled antibodies and 543 nm for Alexa Fluor 568-labeled antibodies. The corresponding fluorescence signals were detected using the emission filters of 425–475 nm, 500–550 nm and 570–640 nm, respectively.

### 2.14. Live-Cell Analysis

Shadoo and prion protein overexpressing cells (Sho-EYFP and PrP-EGFP cells, respectively), the control EYFP and EGFP cells, and parental N2a cells, were transiently transfected by the ORFSpark-tagged calnexin expression plasmid pCMV3-C-OFPSpark. Live cells were imaged typically 48 to 72 h after transfection. Cells that were overexpressing both the red/orange fluorescence emitting CNX-OFPSpark (excitation/emission maxima of 549 nm/566 nm), and EYFP or EGFP proteins were selected and imaged to study the localization of the proteins in the fine endoplasmic reticular membranes. Parental N2a cells transfected only by plasmid encoding CNX-OFPSpark were also imaged to test for any morphological difference between single or double transformant cells. Images were acquired with a VisiScope CSU-W1 spinning disk confocal microscope using 100× oil immersion objective and the same setup of lasers and filters as in immunolocalization experiments.

### 2.15. In Vitro Pull-Down Assay

An in vitro pull-down assay was carried out on either the total cell lysates of transgenic PrP-EGFP, control EGFP and parental N2a cells or separately on each of the gradient fractions obtained from PrP-EGFP cells during fractionation. In the case of total cell lysates, samples of 1 mg of total protein while, for fractionated samples, 50 µL of each fraction were incubated with 30 µL of pre-equilibrated Ni-NTA beads in a 1 mL final volume in Tris-sucrose buffer (20 mM Tris-HCl, pH 7.8 and 250 mM sucrose) at 4 °C overnight in 1.5 mL microfuge tubes, rotating head-over-tail. On the next day, beads with the attached proteins were collected by centrifugation at 14,000 rpm for 1 min and were washed by Tris-sucrose buffer containing 0.25% Triton X-100 five times before being analyzed on SDS-PAGE. The proteins bound to the bead along with its binding partners were detected by Western blotting. The binding of prion protein and calnexin was confirmed with four independent experiments.

### 2.16. Co-Immunoprecipitation

Co-immunoprecipitation was performed in the case of FLAG-tagged Shadoo (Sho-EYFP-FLAG-GPI_(Sho)_ protein construct) expressing Sho-EYFP-FLAG cells to look for calnexin binding using either total cell lysates or isolated raft- and non-raft-type membrane fractions. In the case of the membrane-fractionated cell sample, the collected fractions were first subjected to Western blotting to confirm the presence of the target and the various marker proteins and to discern raft and, non-raft fractions. Based on the flotillin-1 and transferrin receptor’s presence along the gradient fractions, all true raft fractions and, separately, all non-raft fractions were pooled to yield a total raft and total non-raft membrane sample. The total protein amount was estimated through SDS-PAGE and densitometry analysis by ImageJ software. Equal total protein amounts (1 mg) of either total cell lysates or pooled (total raft or total non-raft) fractionated samples were subjected to immunoprecipitation by adding 30 µL anti-FLAG affinity beads to the samples in 5 mL of the incubation buffer (50 mM Tris-HCl, pH 7.4, 150 mM NaCl and 1% Triton X-100) and rotating head-over-tail overnight at 4 °C. Next, the beads were pelleted and washed with wash buffer (50 mM Tris-HCl, pH 7.4, 150 mM NaCl and 0.25% Triton X-100) five times. Equal volumes of the samples from a 2× sample buffer were added to the samples, beads with the attached proteins, to proceed with SDS-PAGE and Western blotting. As a negative control, a sample containing only beads was processed in parallel. Immunoprecipitation was carried out a minimum of four times as independent experiments.

## 3. Results

### 3.1. Fluorescent Protein-Tagged Shadoo and Prion Proteins Are Expressed and Localize as Expected in the Transgenic N2a Cells Developed

We chose mouse neuroblastoma Neuro-2a (N2a) cells as a model system, since they had widely been used in the past for the study of the biology and conversion of PrP^C^ [79], as well as for mapping the interactome of prion paralogs [21] and, therefore, are best-characterized for prion studies. To monitor the two prion family proteins, Shadoo (Sho) and prion protein (PrP), we developed N2a cells stably expressing either of these proteins in fusion with a fluorescent protein tag (EYFP for Sho and EGFP for PrP) at their C-termini, preceding their GPI-anchor signal peptides (termed Sho-EYFP and PrP-EGFP cells, respectively). As controls, we developed N2a populations in a similar manner, but stably expressing the fluorescent proteins only, equipped with the signal sequences of the corresponding ER- and GPI-signaling peptides of Sho and PrP: EYFP-GPI_(Sho)_ and EGFP-GPI_(PrP)_ proteins (EYFP and EGFP cells, respectively).

To test whether the tagged proteins and their controls are expressed and localized correctly in the transgenic cells developed, we first inspected the cells by live-cell confocal microscopy imaging (Figure 2A and Appendix A). Both proteins, Sho-EYFP and PrP-EGFP, predominantly localize to the plasma membrane of the cells and to the perinuclear region, where they manifest as intense fluorescent patches in the cytoplasm close to nuclei (Figure 2(Aa–Ad)) and Figure 2(Ae–Ah)), respectively). 

Using a fluorescent marker for the Golgi apparatus (CellLight™ Golgi-RFP), these patches can be identified as the Golgi apparatus (GA) to which the fluorescent protein-tagged Shadoo and prion proteins colocalize (Figure 2(Aa–Ah)). Similar localization patterns are observed in the control cells where the fluorescent proteins possess the GPI anchors of Sho or PrP proteins (Appendix A). These results are in line with previous observations for the localization of transfected and overexpressed PrP and/or Sho in N2a and other cells [47,80,81].

Western blots of the total cell lysates of Sho-EYFP and PrP-EGFP cells confirm the expression of Shadoo and prion protein transgenes when tested by either α-Sho or α-GFP antibodies for the Sho (Figure 2B, lanes 4–6) and by either α-PrP or α-GFP antibodies for the PrP protein construct (Figure 2C, lanes 4–6). The proteins are detected at their expected molecular weights of approximately 45–49 kDa for Sho and approximately 60–70 kDa for PrP—corresponding to fusion constructs with the fluorescent proteins and accounting for possible glycosylations and GPI-anchor addition. In the case of Sho, two bands appear at around the expected weight of Sho-EYFP (~45–49 kDa), recognized by both α-Sho and α-GFP antibodies in Sho-EYFP cells, which are absent in parental N2a cell samples (Figure 2B, lanes 1–2 and 4–5). It should be noted that there are no unequivocally good anti-Shadoo antibodies available and that cross-reactive bands are frequently apparent on blots of various cells by anti-Sho antibodies [47,60,63,64,82]. However, since α-GFP shows similar bands, these might be two forms of Sho. To test if the proteins possess complex N-glycosylations, cells were treated by the PNGase F enzyme, which removes complex N-glycans. Sho and PrP possess one and two sites, respectively, where complex N-glycosylation may occur. As a result, endogenous and untagged overexpressed PrP and Sho are observed mostly as multiple bands corresponding to coexisting proteins with different states of glycosylation. For the fusion proteins in our experimental setup, we see two bands for Sho but do not resolve separate bands for PrP when untreated. However, as a response to enzyme treatment, a small shift is observed for Sho, for both of the bands (marked by arrowhead), which is detected by both α-Sho and α-GFP antibodies—further confirming their identity as Sho-bands and that both forms are glycosylated (Figure 2B, lane 6). The presence of an intermediate-glycosylated form of Sho, of ~16 kDa, which was also sensitive to PNGase F, was identified previously by Pepe and coworkers in the ER of GT1 cells along with the fully (~22 kDa) and the unglycosylated forms (~14 kDa) [76]. The two bands seen here by us could correspond to these two glycosylated forms of Sho, when tagged by EYFP in our N2a cells. The levels of endogenous Sho in N2a cells are known to be below detection limits by Western blot of the cell lysates [21,65] and as also observed by us (data not shown). Contrarily, the endogenous PrP^C^ is well detected in parental N2a (Appendix A) and in transgenic PrP-EGFP (data not shown) cells and appears as multiple bands based on its different glycosylation states ranging between 26 and 42 kDa (Appendix A, lanes 1 and 2), as commonly observed [75,83]. These bands are not well resolved in the case of the PrP-EGFP fusion protein detected here at around 60 kDa by α-PrP and α-GFP antibodies (Figure 2C, lanes 4 and 5). Upon deglycosylation by PNGase F, in the case of endogenous PrP^C^, a clear shift to approximately 25 kDa can be observed for the upper bands of PrP^C^ (Appendix A, lane 3), whereas for the PrP-EGFP fusion protein, only a small shift (of the upper edge of the band) to lower molecular weights is apparent (Figure 2C, lane 6), indicative of PrP being glycosylated when fused with EGFP. For a GFP-tagged PrP, small shifts in 5–10 kDa were reported upon PNGase F treatment in brain homogenates of transgenic mice [84]. Our result is in line with such a range. The FACS analysis indicates that more than 97% of cells express the protein construct in the Sho-EYFP and PrP-EGFP transgenic populations (Figure 2D). The expression of the control proteins in the transgenic N2a/EYFP and EGFP cells are also confirmed by Western blotting, and the GPI-anchored control proteins run at their expected molecular weights at approximately 30 kDa (Appendix A).

### 3.2. While Sho and PrP Are Membrane-Raft Localized, They Are Present in Non-Raft Membrane Fractions as Well

GPI-anchored proteins, such as PrP and Sho, are secreted to the plasma membrane and are commonly known to reside in lipid-rafts [47,75], where they participate in specialized signaling processes [85]. Since PrP and Sho were reported to engage in not only similar but also different activities and knowing that functionally different GPI-anchored proteins can be organized in different domains in neuronal cells [86,87], we set forward to analyze their membrane-raft partitioning using the transgenic N2a cells developed and characterized above. To exclude eventual detergent-based artifacts on raft organization [88], we chose the detergent-free method for raft separation of Macdonald and Pike (2005) [78]. In this method, the cells are shared in the absence of detergents and the total post-nuclear fraction of the cell lysate is fractionated further on an OptiPrep continuous density-gradient by ultracentrifugation, resulting in the separation of membrane domains based on their buoyancy. Collecting equal volume fractions from top to bottom of the gradient after centrifugation, the method had been shown to yield clear separation of raft- and non-raft-type membrane domains [78]. To analyze the distribution of Sho and PrP, we processed samples of Sho-EYFP and PrP-EGFP transgenic cells in parallel and their control, EYFP and EGFP cells, respectively. To determine the characteristic densities of the collected gradient fractions within our experimental settings, we calculated the densities via measuring the absorbance of each fraction from multiple blank (similar but without the protein sample) OptiPrep continuous density gradients, which were prepared and centrifuged in parallel with the samples (Appendix A). The fraction volumes were collected similarly for the blank and the samples throughout all experiments. The mean density values obtained for the fractions increased from 0.944 g/mL (top of the gradient) to 1.356 g/mL (bottom of the gradient). The small standard errors obtained from measuring three separate blank density gradients in individual experiments show the remarkable reproducibility of the gradient fraction densities from experiment-to-experiment (Appendix A). To analyze the type of membrane domains, we characterized the collected fractions for total protein and cholesterol content and the presence of various proteins (Figure 3).

In order to discern which of the gradient fractions are the most reflective of a lipid-raft-like environment, we used the criteria followed by Persaud-Sawin et al. (2009) [89], according to which fractions that possess low protein content and high cholesterol content show the presence of flotillin-1 (which is known to be abundant in rafts) and the absence transferrin receptor (TfRC) (which is considered as a non-raft resident protein) are identified as raft-fractions. For easier assessment of the protein distributions, we classified the gradient fractions into three density classes: low-density (fraction numbers 1 through 8), mid-density (fraction numbers 9 through 12) and high-density (fraction numbers 13 through 18) fractions. The total protein amount profiles (Figure 3A,B, green solid- and dashed lines) across the gradients of Sho-EYFP and PrP-EGFP (Figure 3A), and the control, EYFP and EGFP (Figure 3B) cells show very low amounts in the low-density fractions and low amounts in the mid-density fractions, while the bulk of the proteins is found in the high-density fractions. Measuring the total cholesterol content of the same fractions, high cholesterol amounts are typically found in the low- and mid-density fractions, with the highest being in fraction #4, and they decline towards high-density fractions #12 through 18, which is contrary to the total protein amounts (Figure 3A,B, purple solid- and dashed lines). Furthermore, using Western blot analysis, we followed the relative distributions of flottilin-1 (a known raft resident protein) and transferrin receptor (as a known non-raft protein) in the collected fractions from the four types of cells (Figure 3C–F). In general, flotillin-1 distributes across the entire gradient from the bottom to the top (high- to low-density fractions), in line with earlier observations [78,90,91], while transferrin receptor, which is known as non-raft resident protein [92,93,94], is mostly retained in high-density fractions, starting from fractions 12 through 18. The “true raft” criterion was analyzed for each sample, but in general, beginning from the top low-density fraction through to the 11th fraction of the mid-density region, the raft criteria was fulfilled.

Besides these proteins, we also blotted for calnexin (CNX), a marker protein of endoplasmic reticulum, and for the nuclear pore complex protein (NPC), a marker of the nuclear envelope membrane, along with the target proteins PrP, Sho and control protein constructs.

When analyzing the target proteins, PrP is seen distributed across all density gradient fractions of PrP-EGFP cells (from fractions 1 through 18), spanning through both raft and non-rafts, while showing higher amounts in the mid-density raft-fractions compared with the low-density raft-fractions, as detected by both α-PrP and α-GFP antibodies (Figure 3C). Flottilin-1 shows a similar distribution to that of PrP, and the non-raft plasma membrane marker TfRC distributes from fraction numbers 12 through 18, marking these as non-raft type fractions. A similar distribution pattern is observed also for the untagged PrP in the corresponding overexpressing transgenic stable PrP(+EGFP) cells (Appendix AA) and for the endogenous PrP^C^ in mother N2a cells (Appendix A), indicating that occupying both kinds of environments is a natural characteristic of the protein. This also indicates that the addition of the fluorescent protein tag (EGFP) and/or the overexpression of the protein did not affect the natural partitioning of PrP within membrane microdomains. The control protein EGFP-GPI_(PrP)_ monitored by the α-GFP antibody in the EGFP cell samples shows a similar distribution, spanning the raft and non-raft fractions (Figure 3D), indicating that this type of distribution is not PrP-specific. Probing for calnexin by α-CNX, this protein shows a distribution essentially similar to PrP, starting from low-density raft-fractions through non-rafts, being more abundant in the mid- and high-density fractions of both cells (Figure 3C,D). The nuclear pore complex protein can only be detected in the high-density fractions and in low amounts, usually from fractions 12 through 18 in both cell samples (Figure 3C,D).

Fractionating the Shadoo-expressing Sho-EYFP and its control EYFP cells, similarly to that in the prion protein-expressing cells, we analyzed the distribution of Sho in parallel to its control protein (EYFP-GPI_(Sho)_) by Western blotting the gradient fractions (Figure 3E,F). When probed by either the α-Sho or α-GFP antibody, Sho appears present in the low-density fractions 2 through 8, although in relatively low amounts, being more abundant in the mid-density raft-fractions (from 9 through 12) while present also in the non-raft type fractions—as distinguished by flotillin-1 present across all fractions and TfRC detected from fractions 12 through 18 (Figure 3E). This distribution of Sho is qualitatively similar to that of PrP (Figure 3C and Appendix A). Calnexin is detected from fraction numbers 3 through 18, residing in both raft- and non-raft-type membranes. NPC is absent in the low- and mid-density fractions and is only detected in the high-density non-raft fractions, where unspecific bands are also detected by the α-NPC antibody (Figure 3E,F). These marker proteins have the same distributions in the fractions obtained from the control EYFP cells (Figure 3F). The control protein EYFP-GPI_(Sho)_ across the gradient fractions of EYFP cells is detected from fraction number 3 by α-GFP antibody (Figure 3F), showing a similar distribution to that observed for the Sho, PrP and EGFP-GPI_(PrP)_ protein constructs (Figure 3C–E). Furthermore, the distribution of EYFP-GPI_(Sho)_ matches the distribution of flotillin-1 through these fractions.

Taken together, these results indicate that the two prion family proteins have similar preferences for membrane microdomain partitioning in these cells and that both proteins reside not only in rafts but also in non-raft-type membrane domains, which is not different from a GPI-anchored fluorescent protein, possessing GPI signal sequences of either PrP or Sho.

### 3.3. Shadoo and Prion Proteins Colocalize with Calnexin in the ER Compartments of N2a Cells

In our membrane-raft fractionation experiments, calnexin is monitored in parallel to PrP and Sho and it appears detectable in both lipid-raft and non-raft membrane fractions of each of the transgenic cells studied (Figure 3). Furthermore, calnexin has been reported to bind PrP^C^ [77]. Therefore, we were intrigued to examine whether this interaction of PrP and CNX is confined to raft- or non-raft-type membranes, and we wanted to test whether CNX is also a binding partner of Sho.

To this, first, we set out to examine the subcellular localizations of calnexin, PrP and Sho, using the transgenic cells developed. Second, we aimed to test their interactions by pull-down or IP-assay. For the latter purpose, we inserted a FLAG tag into the Shadoo-EYFP protein construct, which we engineered between the EYFP and the GPI-signal peptide coding sequences (Appendix A and Materials and Methods), to enable immunoprecipitation experiments. Accordingly, we also developed two additional transgenic N2a cell populations: one stably expressing a Sho-EYFP-FLAG fusion protein, and a corresponding control cell population, stably expressing EYFP in fusion with the FLAG tag and flanked by the ER-targeting and GPI-signal peptides of Sho. We termed these cell populations Sho-EYFP-FLAG and EYFP-FLAG cells, respectively. Confocal microscopy and Western blotting confirmed proper subcellular localization as secretory proteins and the adequate levels of expression of these protein constructs in the transgenic N2a Sho-EYFP-FLAG and EYFP-FLAG cells (Appendix A). These were also found for non-FLAG protein constructs in Sho-EYFP and EYFP cells (Figure 2A,B; Appendix A).

To examine the subcellular colocalization of the transgenically expressed Sho and PrP with endogenous calnexin, we first performed immunocytochemistry combined with confocal microscopy on fixed and permeabilized cells of the Sho-EYFP-FLAG and PrP-EGFP cells expressing the proteins (Figure 4).

Using the same antibodies for detection (α-GFP primary combined with Alexa Fluor 488-labeled secondary antibody), the EYFP and FLAG-tagged Sho protein and EGFP-tagged PrP are seen localized to the same subcellular compartments: plasma membrane (PM), ER membranes and Golgi apparatus (GA) (Figure 4b,f), as found also for the non-FLAG constructs earlier by live-cell analysis (Figure 2A). The endogenous CNX shows similar localization in all cells examined, marking the ER membrane network, but were absent from the PM and GA (Figure 4c,g, Appendix A and parental N2a (data not shown)). Examining the co-localizations using the merged fluorescence images, it can be seen that CNX shows partial colocalization with Sho and PrP by immunocytochemistry. Colocalization is confined to the ER membranes, leaving out GA and PM (Figure 4d,h). Within the ER, inspecting Sho and CNX, the yellow pixels showing colocalization are systematically observed along the nuclear membrane, in the tubular ER membrane structures, and in the ER membrane sheets. For PrP and CNX, colocalized signals are also found in all ER areas; however, for both Sho and PrP non-colocalized fluorescence with CNX is also prevalently observed. Partially overlapping localizations for CNX and the protein constructs EYFP-FLAG-GPI_(Sho)_ and EGFP-GPI_(PrP)_ are seen also when examined by immunocytochemistry using the same α-GFP primary and Alexa Fluor 488-labeled secondary antibodies for EGFP/EYFP and α-CNX primary and Alexa Fluor 568-labeled secondary antibody for CNX (Appendix A. In the absence of primary antibody staining, there was only a negligible signal coming from the nonspecific binding of secondary antibodies to the cells, when examined in the case of each cell type (data not shown).

Since the immunocytochemical procedure may not preserve fine details of the cellular structures especially during the membrane permeabilization step and because antibody pairing may not always be ideal for the detection of colocalization, we also opted for a live-cell analysis of Shadoo, the prion protein and calnexin proteins’ localizations. For this purpose, we transiently transfected the stable transgenic N2a cells with a plasmid coding for a red-fluorescent protein-tagged mouse CNX. When the CNX-transfected Sho-EYFP-FLAG or PrP-EGFP cells are visualized live, under a spinning disk confocal microscope, we can observe the fine subcellular structures expressing the proteins Sho, PrP and CNX-RFP (Figure 5).

Here, the Sho and PrP’s fine ER localization to both the perinuclear and the peripheral ER network, including to both the tubular ER and the sheet-like ER cisternae, is observable. These ER compartments are occupied also by CNX, which has a marked presence in the tubular ER structures. In Sho-EYFP-FLAG cells, Sho is clearly seen localized also around the nucleus marking the nuclear envelope membrane perfectly, where it also has a complete colocalization with CNX (Figure 5a–c). Sho is also equally seen localized to the tubular ER and smooth ER sheets, where it also colocalizes with CNX. In the PrP-EGFP cells, PrP is also present in the nuclear membrane, but it does not characteristically highlight the nuclear envelope, and similar to Sho, it is present also in the tubular ER and ER cisternae (Figure 5d). In all three compartments, CNX is also present and apparently has a complete colocalization with PrP (Figure 5d–f). Apart from these three ER compartments, Sho and PrP are present in the GA and PM, where CNX is absent.

The EYFP-FLAG-GPI_(Sho)_ and EGFP-GPI_(PrP)_ proteins are seen localized essentially to the same subcellular organelles as the Sho and PrP in live-cell microscopy when the stable transgenic EYFP-FLAG and EGFP cells are transiently transfected by the same plasmid expressing the red-fluorescent protein-tagged CNX (Appendix A). Their most intense fluorescence is seen in the PM and GA, while the ER compartments, such as nuclear envelope, tubular and sheet-type ER membranes, have less intense and more homogeneous fluorescence with less structured appearance (Appendix A compared with the Sho- and PrP-expressing cells (Figure 5). Nevertheless, in these ER compartments, the control proteins have overlapping localizations with CNX (Appendix A merged images, yellow). The localization pattern of CNX in these cells is similar to its pattern observed in Sho- and PrP-expressing cells and in parental N2a cells.

### 3.4. Interaction of Prion Protein and Shadoo with the ER Chaperon Calnexin in the Lipid-Raft and Non-Raft Membrane Domains

The prion protein was shown to interact with the lumenal domain of calnexin in both in vitro and in vivo experiments [77]. Finding overlapping localizations of calnexin with PrP and Sho, we set forward to investigate their possible interactions. First, we performed pull-down experiments from total cell lysates of prion protein-expressing PrP-EGFP cells in parallel to its control EGFP and of the parental N2a cells, using Ni-NTA beads that naturally bind PrP (Figure 6A). Western blot analysis of the bead-pulled samples (“Bead eluates”) in parallel to the input total-cell lysates (“Cell lysates”) loaded as the control on the same gels show the presence of CNX in the bead eluates of both prion-expressing cells and N2a samples, whereas, in the bead eluate of the control cell sample, there is only a faint band apparent corresponding to CNX, which is in line with PrP^C^ being endogenously expressed in the parental N2a cells (Figure 6A). This also indicates that prion proteins, either endogenous or overexpressed with the EGFP tag, interact with calnexin.

Next, we aimed to analyze if there is any preference for the interaction of prion proteins and calnexin depending on the type of membrane domains that the proteins reside in. During gradient fractionation and Western blotting of PrP-expressing cells (either PrP-EGFP, PrP(+EGFP) or parental N2a cells), we found that PrP and CNX are usually present in the same gradient fractions and that their detection by the antibodies used is also good (e.g., Figure 6B). These made it feasible to perform individual pull-down assays from each fraction. After density-gradient fractionation and Western blot analysis of the collected fractions of the PrP-EGFP cells, we identified the raft-type fractions from those ranging from 1 to 11 and the non-rafts one from 12 through 18, based on monitoring flottilin-1 and TfRC proteins, and confirmed the presence of PrP and CNX along the fractions (Figure 6B). To test for their interaction, we subjected the individual fractions to Ni-NTA bead-pull-down assay. Ni-NTA beads naturally bind PrP through PrP’s OR region, without the need to use an intermediary antibody. Testing the bead eluates for the presence of both proteins PrP and CNX by Western blot analysis of the bead-pulled fractions, we can observe the presence of both PrP and CNX starting from low-density fraction #5 through the mid- and high-density fractions up to #18, indicating that CNX is pulled in both raft and non-raft membranes (Figure 6C). The “only bead” condition was used as a negative control and presented no background signal (not shown), as previously seen for total cell lysates (Figure 6A). We obtained qualitatively similar results in at least four repeated experiments. Overall, these results indicate that prion and calnexin may interact in both raft- and non-raft-type membrane domains.

Furthermore, we explored whether Shadoo would also interact with calnexin. First, we explored their interaction using the total cell lysates of Sho-overexpressing Sho-EYFP-FLAG cells in parallel with those of its control EYFP-FLAG cells and performed co-immunoprecipitation assays using anti-FLAG beads (Figure 7A). Parental N2a cells were also tested in parallel, as a negative control (Appendix A).

As can be seen in Figure 7A, calnexin is present in the Western blots of the anti-FLAG-bead eluate of the sample corresponding to the Shadoo-expressing cell, whereas it is absent in that corresponding to the control cells, even though the direct loading of both total lysates (“Cell lysate”) show the presence of calnexin, as expected. The total lysates of parental N2a cells loaded to anti-FLAG beads did not result in background signals (Appendix A). The endogenous level of Sho expression, contrary to the prion protein, is undetectable in these cells. These results indicate that the expressed Shadoo protein interacts with calnexin in the cells.

To analyze the binding of Shadoo and calnexin within the raft and non-raft membrane domains, we fractionated the membranes from Sho-EYFP-FLAG and its control EYFP-FLAG cells, using identical procedure as earlier, for the non-FLAG-tagged protein expressing Sho-EYFP and EYFP cells. Western blotting the collected fractions of the two cells in parallel, we can see that the distribution of flotillin-1 and transferrin receptor protein through the fractions are similar for the two cells, (and analogous to their distributions in the fractions of non-FLAG-type Sho-EYFP and EYFP cells). Therefore, we can identify fractions #1 through 11 as a raft type, whereas fractions #12 through 18 can be identified as a non-raft type; for both cells, the separation between them is marked by the start of the overlap of TfRC with flottillin-1 (Figure 7B). We found Shadoo to be present starting from fraction #5, when probed by α-GFP, or from #9, when probed by α-Sho and α-FLAG antibodies, through fraction #18, occupying both raft and non-rafts. This corresponds to that seen for Sho in the case of the non-FLAG type cells, mostly being present in the mid- and high-density fractions (Figure 3E), although the ratio of Sho in the non-raft fractions (#13 through 18) here is somewhat higher compared with the mid-density fractions, which was not observed for the non-FLAG-tagged Sho-expressing cells. Calnexin distributes similarly through raft and non-raft fractions in both Sho-expressing and control cells, appearing from fractions #4 through 18. The control protein, EYFP-FLAG-GPI_(Sho)_ in the control cells is seen to distribute over fraction #5 through 18, with a somewhat wider density region compared with Sho-EYFP-FLAG, as probed both by α-GFP and α-FLAG antibodies.

Since the Western blot signals for Sho (as detected by α-Sho or α-FLAG) are generally weaker in the fractions belonging to the low-density region (Figure 7B) compared with signals obtained by α-PrP for samples of PrP-expressing cells (Figure 3C and Figure 6B), in order to test the Sho and CNX interaction in the different membrane domains, we considered it more feasible to pool all raft-type fractions and to pool all non-raft fractions to form two samples rather than using individual fractions for performing co-immunoprecipitation. Samples of equal total protein amounts from the pooled raft and non-raft samples of Sho-EYFP-FLAG and EYFP-FLAG cells were loaded to anti-FLAG beads to immunoprecipitate the FLAG-tagged proteins. When the bead-eluted samples were analyzed by Western blotting, we can see that both Shadoo (by α-Sho, α-FLAG and α-GFP) and calnexin (by α-CNX) are present in both raft and non-raft samples of Sho-EYFP-FLAG cells (Figure 7C). In the similarly pooled fractions of the control, EYFP-FLAG cells, even though the control protein is present as detected by α-FLAG and α-GFP, CNX could not be detected in the bead eluates consistently, in either rafts or non-rafts. Since we found that, in the case of the raft-fractionated samples, the detectable Co-IPed calnexin signal in the Sho-overexpressing cells was generally not strong (Figure 7C), we considered careful evaluation against the control sample over repeated experiments (at least four) and using different exposures. We also noticed that, in some cases, we picked up faint signals in control cells for calnexin, which were however not consistently repeatable and appeared to also be much less than that for the Sho-expressing cells; thus, we deem it to be an unspecific signal. Taking together, in both total cell lysates and fractionated raft and non-raft membrane samples, we found that calnexin Co-IPes with Shadoo. This indicates that an interaction between Shadoo and calnexin is likely in place, and similar to prion protein, this binding apparently occurs in both the lipid-raft and non-raft membrane environments.

## 4. Discussion

By using transgenic N2a cells expressing prion or Shadoo proteins tagged by EGFP or EYFP at C-terminals preceding their GPI-signals and by applying a non-detergent-based membrane raft fractionation method, we found that, when GPI-anchored Sho and PrP are present in membrane microdomains called rafts, they partition into both raft- and non-raft-type membranes.

Lateral domain formation [95], or rafts, were long-debated for their physiological existence in cell membranes due to a lack of direct proof. Although at present, this still awaits technological developments [96], unfolding breakthroughs in, for example, the resolution limits of microscopy techniques, may likely provide such a possibility in the near future. Together with the expanding knowledge on the attachment and action of the actin cytoskeleton on the plasma membrane pointing to its active role in defining membrane domains and location of GPI-anchored plasma membrane proteins, which also may attach to it [97], this may lead to exciting new insights to reshape our current understanding of rafts. Nevertheless, the existence of membrane rafts is widely accepted as valid, based on indirect evidence gathered through various imaging, biochemical or analytical approaches or computer modeling [98]. Rafts are currently defined as dynamic membrane microdomains enriched in cholesterol, glycosphingolipids and phospholipids acylated with saturated fatty acids, which form ordered-lipid platforms that confine temporarily or stably key proteins to perform specific signaling activities and, hence, compartmentalize in this manner cellular processes [99,100]. In vivo, rafts are postulated to be small, heterogeneous in size (~10–200 nm) and composition, and highly dynamic in nature. They laterally move, temporarily disband to smaller, or join into larger platforms stabilized through protein–protein, protein–lipid or lipid–lipid interactions [101]. Recent neutron scattering data provided for the first time in situ evidence of ~40 nm size domains in functional bacterial membranes [102]. Among the indirect biochemical approaches, the classical detergent-based (e.g., 1% TritonX-100; CHAPS; Brij 58, 96 and 98; Lubrol; Nonidet P40; octylglucoside; and others) cell fractionation followed by sucrose density-gradient separation led to the isolation of “detergent-resistant membrane fractions” (DRMs) with the attribution of the membrane rafts being the low-density floating fractions. Although not equivalent to in situ rafts, their characterization had largely contributed to our current understanding of rafts in mammalian cells [103,104]. These techniques are widely used and have also been applied to the study of prion proteins [47,76,105]. Several observations, however, raised concerns over this method, pointing to the ability of detergents to cluster lipids and proteins originally not associated with each other into a raft fraction. The high variation in the reproducibility of this method is also argued to be the reason for this experimentally induced clustering effect of detergents [88,106,107]. Hence, non-detergent-based methods have been developed [89,108]. In line with these, we opted for using a non-detergent-based raft-isolation method, one improved by Macdonald and Pike in 2005 [78]. The approach is known to yield high-purity raft separations by shearing the cells in detergent-free conditions and fractionating the post-nuclear supernatant on a continuous OptiPrep density-gradient yielding separation of rafts and non-rafts based on density flotation. The low-density fractions occupied by rafts on this gradient correspond to the densities occupied by rafts floating over sucrose density gradients following the detergent-based extractions [78].

PrP and Sho tagged by EGFP and EYFP (or EYFP-FLAG), respectively, or the untagged PrP are correctly processed when stably expressed in the N2a cells developed here. They are complex glycosylated and show subcellular localizations as expected for GPI-anchored corresponding to maturing through the secretory pathway in the ER, Golgi apparatus and PM (Figure 2, Appendix A). This is also in line with reported localizations of PrP and Sho transgenes in different cells [47]. Fractionating the cell membranes and separating the raft- and non-raft-type domains confirmed that these proteins, occupy low-density fractions (Figure 3 and Figure 7). These fractions correspond to raft-type membranes, according to the criteria for “true raft-fractions” of Persaud-Sawin and coworkers (having low protein and high cholesterol content and showing the presence of flotillin-1, as a protein abundant in rafts, and the absence of the transferrin receptor protein as a typical non-raft protein) [89]. This kind of partitioning of PrP and Sho to rafts is in accordance with previous observations based on employing detergent-based raft fractionation methods [47,76,105]. Interestingly, while PrP and Sho are present along the raft-type membrane fractions, they reveal a preference for residing in the mid-density raft-region (fractions #9 through 12); moreover, they are also detected in the high-density region, where typically, the non-raft marker transferrin receptor is detected and, thus, are considered non-raft fractions (Figure 3C,E and Figure 7B, Appendix A). This pattern for the partitioning of Sho and PrP^C^ was consistent over repeated experiments. The separation of raft-type fractions from non-rafts consistently (in repeated experiments and across the different N2a cells examined) also noteworthily occurs at approximately fraction numbers 12–13, marked by the appearance of TfRC, which provides a good resolution for rafts and non-rafts and reflects the remarkable reproducibility of the separation method.

Studies indicated that functionally different GPI-anchored proteins can be organized into different types of rafts [86] and that structural aspects of both the anchor and of the protein can influence their localization to distinct membrane environments [87,109]. In this respect, the anchor signal sequence, as well as the final composition of the anchor, differs for Sho and PrP in that PrP’s anchor possesses rare sialic acid modification. This was referred to as a major determinant of PrP occupying the basal membranes of the polarized MDCK cells as opposed to the apical side [110]. Similar observations were made for PrP and Thy1, a major neuronal GPI-anchored glycoprotein, where PrP was found to reside in different rafts, namely in “more soluble rafts” (as compared to Thy1), which also harbor different compositions of proteins enclosed within [86]. However, here, we could not observe a clear difference between the gradient distributions of PrP and Sho despite them possessing different GPI-signal sequences and despite using a non-detergent-based raft fractionation, which is known to better preserve the natural membrane domain milieu. This is recapitulated also by their control proteins EGFP-GPI_(PrP)_ (Figure 3D) and EYFP-GPI_(Sho)_ (Figure 3F) with GPI-signal sequences of PrP and Sho, respectively, which follow distribution patterns similar to each other and to the prion protein (Figure 3C) or Sho (Figure 3E). This indicates that the GPI-signal sequence difference alone does not assign an overall differential localization for Sho and PrP as viewed through these membrane domains and techniques used.

The non-raft residency of PrP observed here is in line with earlier observations made based on detergent-based raft fractionation of brain neuronal cells or N2a cells [86,111]. In the case of N2a cells, nearly 50% of PrP is shown to be in non-rafts [111]. Additionally, it is in line with the observations that PrP (endogenous or overexpressed in N2a cells) moves outside the rafts (without or as a response of Cu^2+^ binding to the OR region, respectively) and subsequently participates in clathrin-dependent endocytosis via its N-terminal polybasic segment together with transferrin receptor [105,111]. Here, we also report that a large fraction of Sho, similar to PrP, also resides in non-rafts. However, the trafficking of Sho in and outside rafts and its underlying activity have not yet been demonstrated. Notably, however, several plasma membrane proteins have been observed to be able to reversibly join or move out of rafts during their activity [112]. An indication for a similar situation and possibility for Sho may be inferred from the interactome analysis of the prion-family proteins by Watts and coworkers, showing that proteins with oligomannosidic N-glycans such as the transferrin receptor [21,113]—the non-raft marker protein used here—specifically co-purified with PrP and Sho in N2a neuroblastoma cells as well as in wild type mice brain [21]. The oligomannosidic modification is found on a small number of proteins typically found in the brain, such as cell adhesion molecule L1, integrins, nucleotide pyrophosphatase-5 and the b-subunit of Na/K ATPase [114,115,116]. These results suggested that these proteins populate similar microenvironments with PrP and Sho, although, as the authors reasoned, they likely have indirect binding to PrP and Sho via possibly NCAM and/or basigin, known to bind both oligomannosidic N-glycans and PrP^C^ [117,118].

To examine the CNX binding of PrP and Sho using the transgenic N2a cells expressing the proteins, first, we tested the colocalization of the proteins with CNX by both immunocytochemistry and live-cell analysis. CNX is mostly known to function in the ER as a chaperone. The ER is an extensive cellular compartment composed of the nuclear envelope and the peripheral ER network, where the peripheral ER is a complex structure further comprising interconnected tubular networks and flat sheet-like cisternae [119,120,121]. The ER maintains a dynamic ratio of the two tubular and sheet-like structures via specific proteins that are involved in shaping the ER, possibly in accordance with the cell’s needs for the different processes taking place in these different structures [122,123]. By both immunohistochemistry of the endogenous and live-cell analysis of transiently transfected CNX, its localization showed partial overlap with that of PrP and Sho in the cellular organelles, being absent in the PM and Golgi apparatus, confirming its in-bulk retainment in the ER. When visualizing the proteins in the live cells stably expressing PrP or Sho and transiently expressing the fluorescent-protein tagged calnexin, the fine ER structures are better seen and localizations are monitored in a more natural condition, compared with immunohistochemical specimens. In live cells, both Sho and PrP appear to be colocalized with calnexin in all three ER compartments. Sho is clearly present in the nuclear envelope, where PrP is also present but less intense and where they colocalize with CNX. Additionally, Sho and PrP, together with CNX, are detected colocalized in the tubular ER and the sheet-like ER membrane structures as well (Figure 5). It is notable that these fine colocalization patterns were not detectable in immunocytochemistry, which may be attributable to the sample preparation involved, specifically fixation and permeabilization, by which the fine subcellular morphologies of the ER may have not preserved but which are seen in the live-cell imaging. In the case of the live analysis of control proteins in the EGFP and EYFP-FLAG cells transiently transfected by CNX, we see that, while the proteins share overlapping localizations only in the ER, we observed less pronounced yellow fluorescence on the merged images, indicating possibly partial colocalization of the control proteins with CNX within the ER tubular network compared with PrP and Sho (Appendix A).

When testing the binding of PrP and Sho with CNX, first, we employed total cell lysates. For PrP, the Ni-NTA-bead pull-down experiments on the total lysates of PrP-EGFP cells confirm the presence of CNX in the bead-pulled PrP-containing eluates (Figure 6A), confirming that CNX may bind PrP, in line with earlier observations by Wang and coworkers [77]. Performing pull-down assays of PrP in each of the 18 separated membrane fractions, we report here that PrP pulls CNX in both raft and non-raft membrane fractions (Figure 6C), which indicates that at least a portion of PrP may be bound to CNX irrespective of PrP being in the rafts or non-raft type domains. In the case of Shadoo, to test its binding to CNX we developed cells stably expressing FLAG-tagged Sho-EYFP (Sho-EYFP-FLAG cells) and used anti-FLAG-beads for immunoprecipitation of Sho. Using the total cell lysates of these cells first, we show here that calnexin Co-IPs with Shadoo (Figure 7A). Separating rafts and non-raft membrane fractions by the non-detergent-based method, we furthermore show that, in the pooled fractions of raft and non-rafts of the fractionated cell membranes, CNX co-immunoprecipitates with Sho in both raft-and non-raft environments (Figure 7C). These data indicate that calnexin accompanies Sho as well, just as PrP, presumably bound to it, in either the raft or non-raft membrane domains.

While predominantly located in the ER, interestingly, both calreticulin and calnexin have been identified at the cell surface of a number of cells [124,125]. Calnexin was found at the cell surface together with glycoproteins and had been proposed to perform chaperoning functions also at the plasma membrane, although by only a small fraction of it, and that its actual PM amount may be regulated by its cycling and the balance between its exocytosis and endocytosis [125]. Watts et al. (2009) reported also that all three chaperones ERp57, calnexin and calreticulin are pulled by the mature prion proteins (PrP, Sho and Dpl) at the cell surface, as baits in the surface-crosslinked samples of the interactome analysis of the prion family proteins [21]. The amount of surface ERp57 and CTR was estimated by these authors to be a small fraction each (1:1000) compared with the majority being in the ER. Intriguingly also, a recent study demonstrated that GPI-anchored misfolded PrP (PrP*) is recycled from Golgi to lysosomes by transitioning first through the plasma membrane [126], arriving at the cell surface in complex with ER chaperones and cargo receptors, where specifically calnexin and TMED10 were found tightly bound to PrP*. About 0.5% of calnexin was reported by the authors to be at the cell surface at steady-state, while about 85% of misfolded PrP* was temporarily sent to PM for a short residency, contrary to properly folded PrPs arriving at PM that resided for log times. This provided short exposure of the misfolded species to the external milieu. The authors found similar results with other misfolded GPI-anchored proteins and suggested that cells may expose misfolded proteins to the cell surface before proceeding for their degradation as presumably a means of cell-to-cell communication of their health status.

Since in our experiments post-nuclear membranes are subjected to density-gradient fractionation, without prior separation of ER and PM, we can assume that individual or pooled raft (and non-raft) fractions contain membrane domains originating from both compartments, in line also with the notion that PrP associates with lipid rafts already in the early secretory pathway in ER [75]. Hence, we cannot refer to the location of the interaction; however, we can assume that it may presumably be present both at the surface of the PM and in the ER membranes.

Since PrP and Sho were able to co-immunoprecipitate CNX in both non-raft and raft fractions of our transgenic N2a cells, it may be speculated that, at any given time and irrespective of their membrane domain localization, at least some fraction of the raft and non-raft resident PrP^C^ and Sho molecules may be in or may transiently undertake partially unfolded states that require chaperone binding. Such an interpretation could also be in line with the findings of Pepe and coworkers [76] that a percentage of Sho is in an aggregated state in primary GT1 and SH-SY5Y cells in normal conditions, which also co-immunoprecipitates calreticulin. From this perspective, CNX may be at least one of the candidate chaperones, perhaps beside calreticulin, that participates in maintaining their proper fold and in both raft and non-raft membrane regions, apparently both at the PM and in the ER. However, further experiments would be needed to prove or disprove such a proposition in our case.

## 5. Conclusions

By applying the non-detergent-based fractionation of the post-nuclear membranes of transgenic N2a cells, we observed that, while both the prion protein and Shadoo occupy raft-type membrane fractions, a considerable proportion of both are present in the transferrin-marked non-raft membrane domains. We propose that their dual raft/non-raft distribution reflects their loose containment to rafts, with these perhaps also contributing to the multitasking ability of the two proteins. We also report that calnexin, an ER chaperone, is shown to bind PrP^C^, also co-immunoprecipitates with Sho. Moreover, both proteins pull down calnexin in both raft and non-raft fractions. Based on these, we propose that calnexin is not only a binding partner of the two prion family proteins but also needed to assist at least a percentage of mature PrP and Sho populations during their normal biology, irrespective of the proteins being in the raft or non-raft type membrane domains.

## Figures and Tables

**Figure 1 membranes-11-00978-f001:**
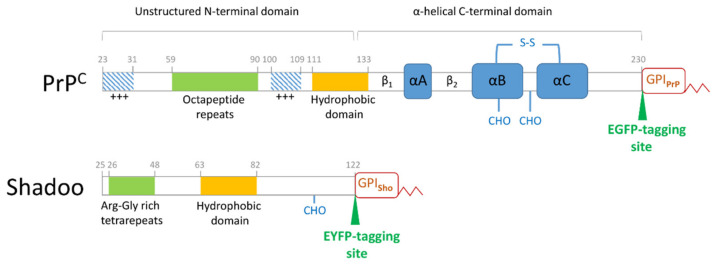
Scheme of the protein sequences of the cellular prion and the Shadoo proteins. The sequence of the native mouse prion protein (PrP^C^) and of the mouse Shadoo are depicted with their characteristic regions marked. The N-terminal unstructured domain of the PrP^C^ contains functional elements, such as an N-terminal polybasic region (+++), the copper-binding octapeptide repeat region, a second polybasic patch (+++) and the highly conserved hydrophobic domain. The globular domain is composed of two short antiparallel β-sheets (β_1_ and β_2_) and three alpha helices (αA, αB and αC) connected by a disulfide bond (S–S). The globular domain harbors glycosylation sites (CHO) at positions 180 and 196 and a GPI-anchor (GPI_PrP_) at the C-terminus. The Shadoo protein is fully unstructured and harbors a conserved hydrophobic domain highly homologous to that of PrP^C^. It possesses also an Arg-Gly tetrarepeat region, one glycosylation site (CHO) at position 107 and a C-terminal GPI anchor (GPI_Sho_). For our studies, an EGFP or an EYFP fluorescent protein was used as a tag to produce the fusion protein constructs PrP-EGFP or Sho-EYFP, respectively, where the fluorescent protein sequences were inserted as indicated on the schemes, at the C-termini, but prior to the GPI-anchor attachment of the proteins.

**Figure 2 membranes-11-00978-f002:**
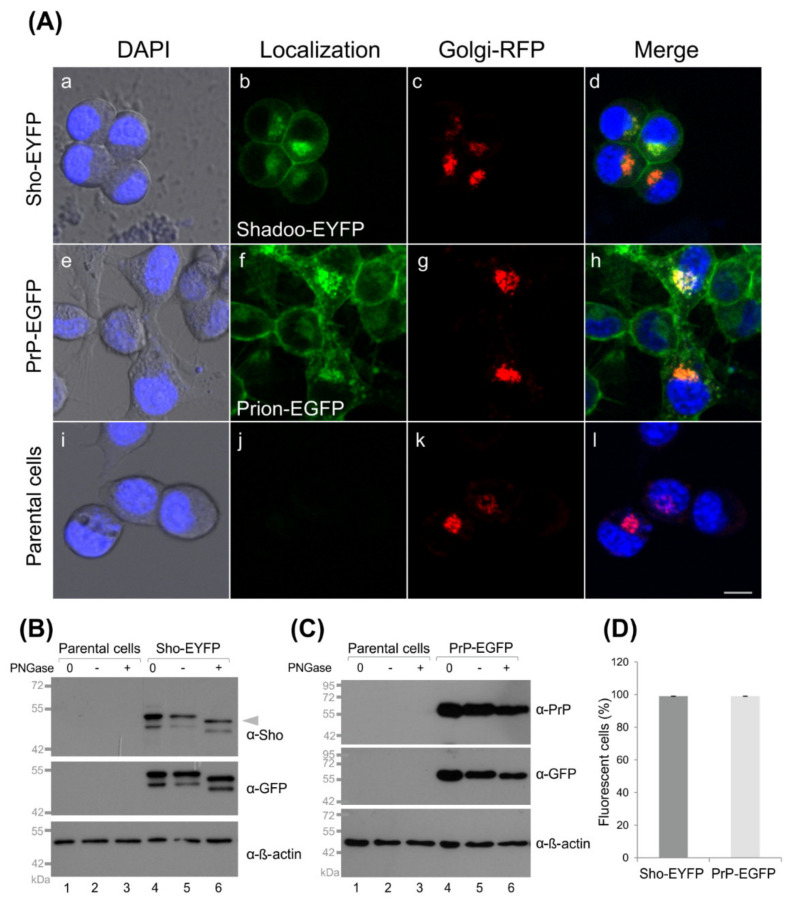
Fluorescent protein-tagged Shadoo and prion proteins are expressed correctly in the stable N2a cells developed. (**A**) Representative live-cell confocal microscopy images of stable transgenic Sho-EYFP (**a**–**d**) and PrP-EGFP (**e**–**h**) cells and of parental N2a cells (**i**–**l**). Overexpressed proteins Sho-EYFP-GPI_(Sho)_ (marked as Shadoo-EYFP) and PrP-EGFP-GPI_(PrP)_ (marked as prion-EGFP) are shown in green. The Golgi apparatus labeled by CellLight™ Golgi-RFP is shown in red. Cell nuclei labeled by DAPI are shown in blue. Transmitted light images are overlaid onto DAPI images in the first column. Scale bar: 10 µm. (**B**,**C**) Western blots of the total cell lysates, of untreated control (0), treated in the absence of PNGase F (−) and treated in the presence of PNGase F (+), of either Sho-EYFP (**B**) or PrP-EGFP (**C**) cells, side-by-side with those of the parental N2a cells. The expression of Shadoo is tested by both α-Sho and α-GFP antibodies; prion protein expression is tested by the SAF-32 anti-prion protein (α-PrP) and by α-GFP antibodies. Note: the same α-GFP antibody is used for recognizing both EGFP and EYFP proteins. The arrow in B indicates the corresponding band for Sho. β-actin is used as a loading control and is tested by α-β-actin. The gels used are either 8% PA (**B**) or 12% PA (**C**) SDS gels. (**D**) The percentage of fluorescent cells as determined by FACS analysis is shown in the transgenic cell populations as marked.

**Figure 3 membranes-11-00978-f003:**
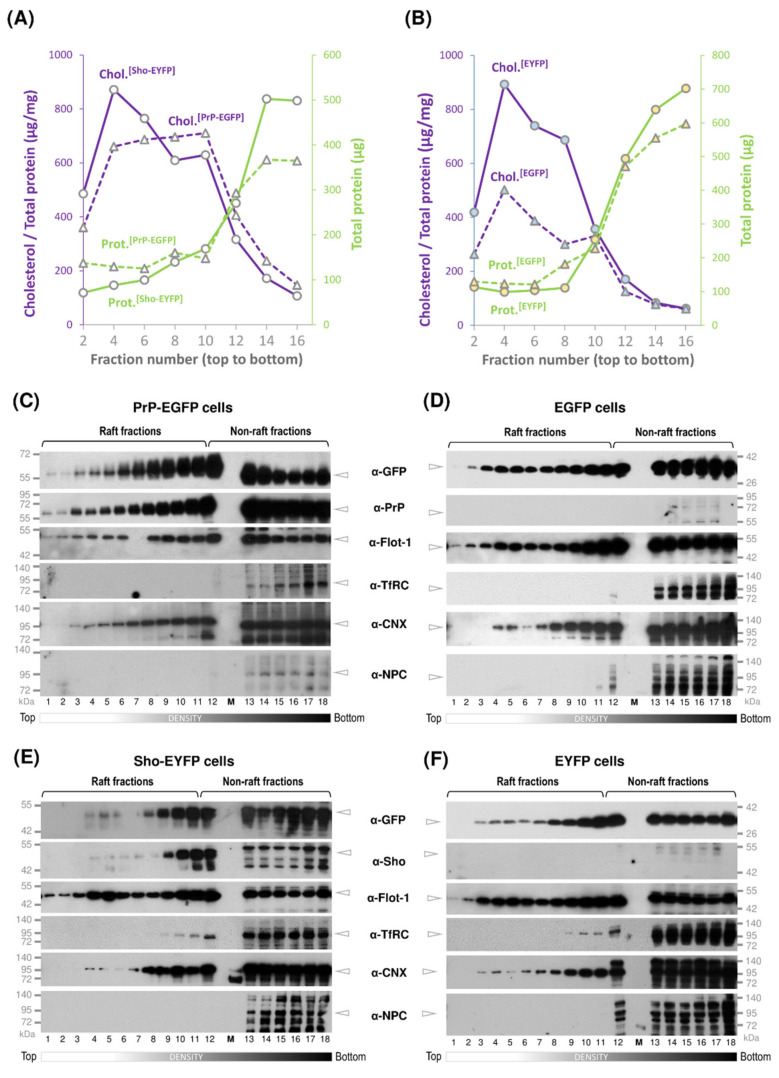
Characterization of membrane fractions and the distribution of the prion and Shadoo proteins across the various density membrane domains obtained by non-detergent based fractionation method. (**A**,**B**) Distribution of total proteins (left *Y*-axis) and cholesterol (right *Y*-axis) in the fractions collected from top to bottom of centrifuge tubes and numbered 1 through 18 of the OptiPrep density gradients of Sho-EYFP and PrP-EGFP cells (**A**) and of their control, EGFP and EYFP cells (**B**). (**C**–**F**) Representative Western blots of the gradient fractions collected from the PrP-EGFP (**C**) and its control EGFP cells (**D**); Sho-EYFP (**E**) and its control EYFP cells (**F**). Fraction numbers are shown below the blots. Samples are immunoblotted for the proteins as indicated. GFP: EGFP or EYFP protein; Flot-1: flotillin-1; TfRC: transferrin receptor protein; CNX: calnexin; NPC: nuclear pore complex protein.

**Figure 4 membranes-11-00978-f004:**
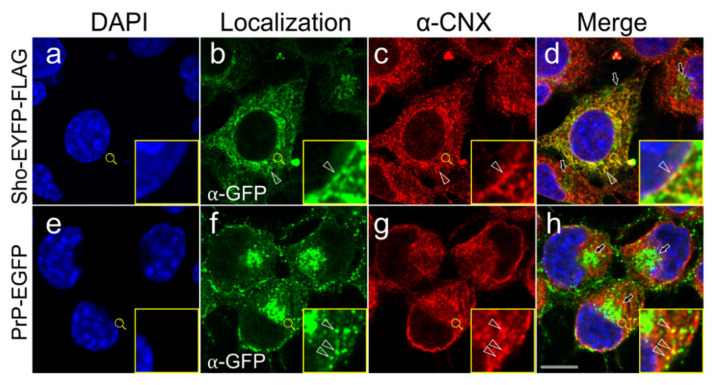
Subcellular localizations of Shadoo, the prion protein and calnexin partially overlap, as detected by immunocytochemistry. Representative immunocytochemistry and confocal microscopy images of stable transgenic Shadoo (Sho-EYFP-FLAG) and prion protein (PrP-EGFP)-expressing cells. Nuclei are stained by DAPI (**a**,**e**). Alexa Fluor 488-labeled secondary antibody is used to detect the primary antibody against EGFP/EYFP ((**b**,**f**); green). Alexa Fluor 568-labeled secondary antibody is used to detect the primary antibody against calnexin, α-CNX ((**c**,**g**); red). Yellow pixels indicate colocalization of Sho or PrP with CNX on the merged images presented in the last column (**d**,**h**). Insets correspond to the areas marked by magnifying glass symbols and highlight representative ER areas for better visualization. Arrowheads mark examples of colocalization areas, whereas arrows point to the Golgi apparatus. Scale bar: 10 µm.

**Figure 5 membranes-11-00978-f005:**
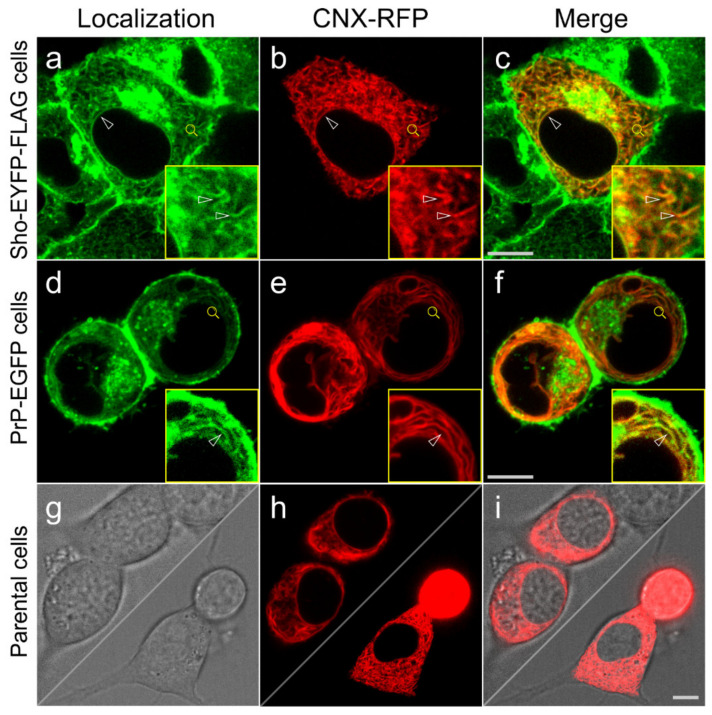
Live-cell analysis shows colocalization of Shadoo and prion proteins with calnexin in the ER membrane network. Stable transgenic Sho-EYFP-FLAG (**a**–**c**) and PrP-EGFP (**d**–**f**) cells expressing EYFP-FLAG-tagged Sho and EGFP-tagged PrP (green), respectively, and parental N2a cells (**g**–**i**) are transiently transfected to express red fluorescent protein (OFPSpark)-tagged calnexin (CNX-RFP, (**g**–**i**), red). Merged images from the green and red channels show the overlapping fluorescence in yellow. Insets show representative ER areas corresponding to positions of the magnifying glass symbols, magnified and with intensity enhancement for better visualization. Arrowheads point to examples of areas with colocalization. Scale bar: 10 µm.

**Figure 6 membranes-11-00978-f006:**
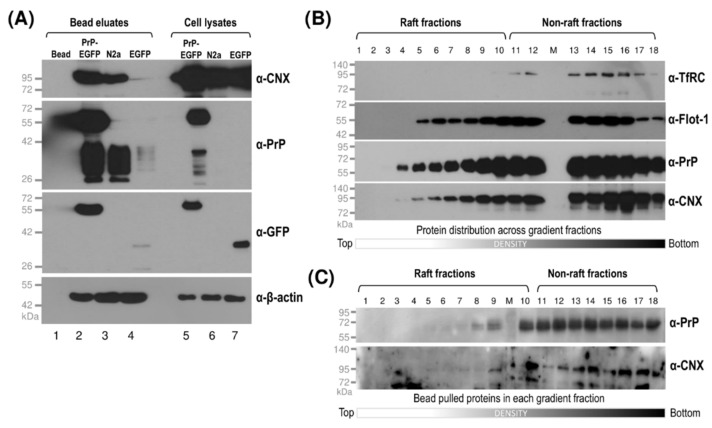
Interaction of the prion protein and calnexin in both raft and non-raft membrane domains indicated by pull-down assay. (**A**) Western blot analysis of the samples after pull-down assay performed on total cell lysates of transgenic PrP-EGFP, EGFP and parental N2a cells. The samples pulled by Ni-NTA beads (Bead eluates) or loaded directly as the input sample (Cell lysates) are probed in parallel for the prion protein (by α-PrP and α-GFP) and for calnexin (by α-CNX). β-actin is used as a loading control and is probed by α-β-actin. Beads, without the sample applied are treated similarly and are used as negative controls (only bead); the molecular weight ladder is indicated on the left side of the blots. (**B**) Separation of raft and non-raft membrane fractions from PrP-EGFP cells to be used for pull-down assay. The fractions are Western blotted for flottilin-1 (α-Flot-1), transferrin receptor, TfRC (α-TfRC), for PrP (α-PrP) and for calnexin, CNX (α-CNX). The fraction numbers collected from top to bottom of the gradient are indicated on the top, and the raft and non-raft density regions are marked. (**C**) Pull-down assay and Western blot analysis of the individual fractions from (**B**). Fractions are pulled by Ni-NTA beads individually and are probed for the prion protein (α-PrP) and for calnexin (α-CNX). M: molecular weight ladder. All gels used are 12% PA SDS gels.

**Figure 7 membranes-11-00978-f007:**
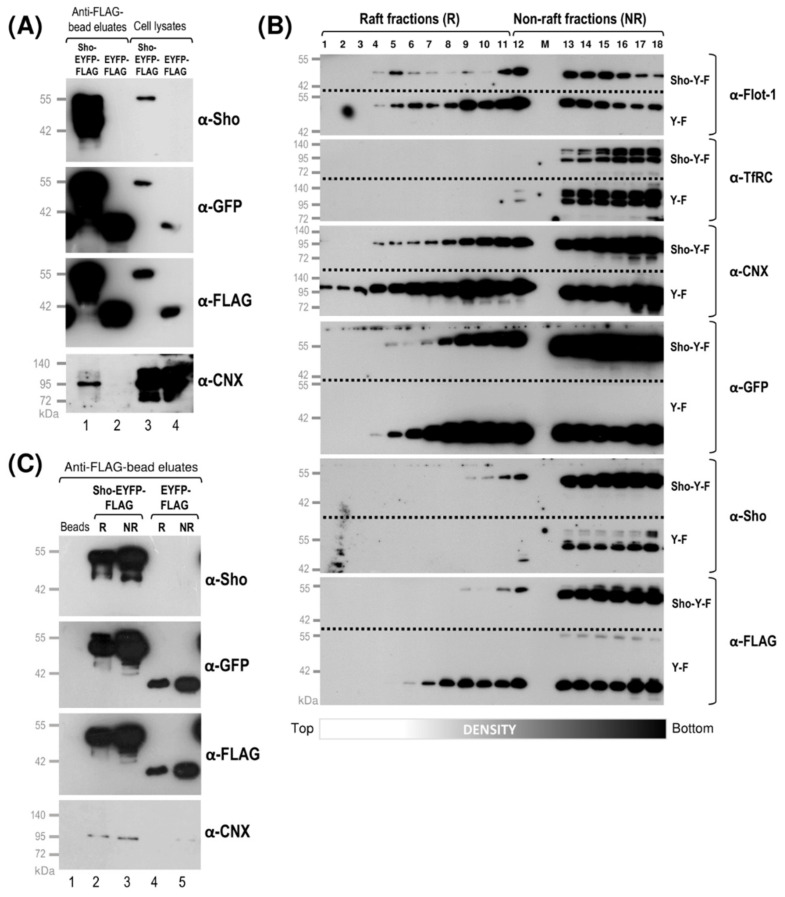
Calnexin is apparently a binding partner of Shadoo in both raft-and non-raft fractions. (**A**) Co-immunoprecipitation of Sho and CNX in unfractionated total cell lysates of Sho expressing Sho-EYFP-FLAG cells using anti-FLAG beads. Bead-pulled proteins (anti-FLAG bead eluates) are tested in parallel to the bead-input cell lysates loaded directly to the gel (cell lysate) by Western blotting for Sho (using α-Sho, α-FLAG and α-GFP antibodies) and for CNX (using α-CNX antibody). Beads treated similarly but without the applied sample are used as a negative control (beads). The molecular weight ladder is indicated on the left side. (**B**) Western blot analysis of the fractions resulting from the OptiPrep density-gradient fractionation of Sho-EYFP-FLAG (marked as Sho-Y-F) and of its control EYFP-FLAG (marked as Y-F) cells. Following SDS-PAGE, the gels corresponding to the two types of cells are cut and arranged one below the other prior to performing a transfer onto the same PVDF-membranes, allowing side-by-side testing for the proteins marked using the antibodies as follows: α-Sho, α-FLAG and α-GFP antibodies for Shadoo; α-CNX for calnexin; and α-flottilin-1 (α-Flot-1) for flotillin-1, and α-TfRC for transferrin receptor. The fraction numbers are indicated on the top, and the raft and non-raft regions are marked. (**C**) Co-immunoprecipitation of Shadoo and calnexin proteins from the density-gradient-separated fractions. Equal amounts of pooled fractions belonging to either raft (R) or non-raft (NR) fractions (as marked on (**B**)) of Sho-EYFP-FLAG or control EYFP-FLAG cells are loaded onto anti-FLAG beads, and the bead-pulled proteins are tested in parallel by Western blotting for Sho and CNX by the same antibodies as in (**B**). All gels used are 10% PA SDS gels.

## Data Availability

The data are included in the article and the Appendix A.

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
