# Peer review of "Membrane Domain Localization and Interaction of the Prion-Family Proteins, Prion and Shadoo with Calnexin"

_membranes, 2021, doi:10.3390/membranes11120978_

Round 1

Reviewer 1 Report

Dondapati et al used GFP/RFP tagged prion and Sho to investigate localization and interaction with  Calnexin and the possible function. The data is well presented and interesting. I only have two comments:

  1. how the interaction of Prion/Sho with calnexin affect PrPsc function?
  2. Any chance to confirm the interaction of prion/Sho with calnexin by immunoprecipatation in animal samples such as mouse brain.
  3. Discuss the limitation.  

Author Response

Replies to the questions and comments raised by the Reviewer 1

We thank the Reviewer for reviewing our manuscript and for providing valuable comments and questions that contributed significantly to the value of the manuscript. We specifically thank for the prompt reviewing and the short time within which we received the comments.

Our point-by-point answers to the questions and comments are written in blue and are as follows.

Reviewer 1:

Dondapati et al used GFP/RFP tagged prion and Sho to investigate localization and interaction with  Calnexin and the possible function. The data is well presented and interesting. I only have two comments:

1.  how the interaction of Prion/Sho with calnexin affect PrPSc function?

Response 1: We do not address directly this question in our MS in the lack of an appropriate model system at hand. There is no available literature data either on the direct effect of calnexin, or of its binding to PrP or Sho, on PrPSc formation. However, one study, by Wang and coworkers, demonstrated that calnexin binds to and decreases the thermal aggregation of PrP in vitro and binds to PrP and decreases the toxicity of PrP overexpression in cultured SK-N-SH cells when transiently overexpressed[1]. Based on these, it may be reasonable to speculate that calnexin may also hinder PrPSc formation, but we lack any of this type of data. This was the reason for not discussing this question in the MS.  However, we refer to and discuss the above article in the context of PrP-calnexin binding in the discussion section.

2.   Any chance to confirm the interaction of prion/Sho with calnexin by immunoprecipatation in animal samples such as mouse brain.

Response 2: We have not performed studies on brain samples, however, we agree with the Reviewer that it is important to look for similar interaction in mouse brain tissue samples, to test the in-vivo relevance of our results. There are no data presented in the literature either, reporting on interaction of PrP and/or Sho with calnexin in brain samples. Apart of our study, there is only one report, of Wang and coworkers[1] mentioned also above, which proves a direct evidence for interaction between PrP and calnexin by both an in vitro and an in cellulo approach. In the latter case, the authors employed co-immunoprecipitation of either endogenous PrPC and calnexin in human SK-N-SH cells, or of the transiently co-expressed PrP and calnexin in 393T cells, using in both cases total cell lysates.  Furthermore, Watts and coworkers[2] conducted an interactome analysis of prion family proteins (PrP, Sho and Dpl), applying native crosslinking using cultured mouse neuroblastoma cells and immunoprecipitation with the prion family proteins as baits followed by mass spectrometry analysis. The authors reported calnexin being among the several other precipitated proteins, which argues for calnexin as a possible binding partner of PrP and Sho (and Dpl) in these cells. The relevance of these data in vivo in animal or human brain tissues remains to be researched.

3.  Discuss the limitation.  

Response 3: In this study we apply transgenic N2a cells and demonstrate the binding of calnexin to PrP and Sho proteins in total cell lysates and also in raft- and non-raft type membrane fractions, separating these domains by a non-detergent based protocol. Although this approach may preserve better the natural organization of membrane domains compared to the widely used detergent-based methods, we have to keep in mind that it still remains an indirect method of study of in vivo, cellular membrane rafts. How the fractionated membranes relate to in vivo highly dynamic membrane rafts, still remains a mystery until future technical developments will allow their direct observation. Other limitation of our results may be that, as the Reviewer also highlights, similar studies on brain tissues would be a further step in proving the relevance of our results in vivo, as well as, it would be intriguing to test if these interactions would play any role in PrPSc formation. In the absence of such studies our data provide limited possibility for inferring on their importance in in vivo situation or in the case of prion disease. 

References

  1. Wang, W.; Chen, R.; Luo, K.; Wu, D.; Huang, L.; Huang, T.; Xiao, G. Calnexin inhibits thermal aggregation and neurotoxicity of prion protein. J. Cell. Biochem. 2010, 111, 343–349.
  2. Watts, J.C.; Huo, H.; Bai, Y.; Ehsani, S.; Jeon, A.H.W.; Shi, T.; Daude, N.; Lau, A.; Young, R.; Xu, L.; et al. Interactome analyses identify ties of PrP and its mammalian paralogs to oligomannosidic N-glycans and endoplasmic reticulum-derived chaperones. PLoS Pathog. 2009, 5.

Reviewer 2 Report

In their manuscript, Dondapati and colleagues use a detergent-free lipid fractionation method to isolate membrane raft and non-raft fractions, which they use to study the localization of the GPI-anchored prion protein PrP as well as its homolog Shadoo. The authors show that the proteins are present in both raft and non-raft fractions. The data is supported by live cell imaging. Finally, the authors show that the two proteins directly interact with calnexin, an ER-chaperone. The authors postulate that the chaperone could be required for the folding of the two proteins, which would subsequently allow them to localize in and out of lipid rafts.

First and foremost I would like to say that I really appreciate the time and effort the authors have put into carefully explaining their methodology. The manuscript really stands out in that matter. Similarly, the results are very nicely presented and structured. This was a very easy manuscript to read and comprehend on.

Below are my (minor) comments for the work:

  1. The English language needs revising, especially in the introduction. Please ask a native English speaker to meticulously correct the manuscript, if possible. Attention should be placed especially on the use of articles. The entire manuscript should be carefully proofread and the language corrected where necessary.

  1. In the introduction, the authors introduce PrP on lines 70 – 87. This is a fairly complicated description and I think the readers of the article would appreciate it if the authors included a figure to illustrate the structure and function of PrP and Shadoo. Additionally, this could enable the authors to draw the domain structures of the protein constructs that are relevant for the study.

  1. I ask the authors to be careful with calling the PrP protein “prion”, as there is some evidence that several different proteins can be considered prions. I know that in the field it is common that “prion” is specifically used to refer to PrP, but an effort should be made to call PrP “prion protein” rather than “prion”. Consider changing this nomenclature for the manuscript.

  1. Perhaps the authors could indicate in Figure 2 which fractions represent rafts and non-rafts, much like in Figure 5. I think this would be helpful to the readers of the article.

  1. The authors refer to Flotillin 1 as a raft marker, but it is clearly present in the non-raft fractions as well. Could the authors maybe explain why this is the case? Is Flotillin 1 really exclusive to rafts?

  1. Since cholesterol is only enriched in rafts, rather than specifically found there, did the authors at any point consider testing for the presence of e.g. sphingomyelin or some other lipid that is known to be specific to rafts or at least very highly enriched in rafts?

  1. I will not ask the authors to perform further experiments, as the work is already convincing as it stands, but I am strongly asking the authors to consider the following, as it would further support their conclusions: a well-established experiment to look at the localization of proteins in lipid rafts is performed using fluorescence microscopy of giant unilamellar vesicles (GUVs). GUVs present raft formation when the membrane lipid composition is correct, and rafts can be visualized in several ways, such as adding fluorescently labelled sphingomyelin (which localizes to rafts) or using fluorescent cholera toxin (which binds to gangliosides that are in rafts). Thus, fluorescent recombinant proteins that localize to rafts should co-localize with the fluorescent signal of the raft markers. If the authors would perform such a cell-free experiment, it would support their already existing data for the localization of PrPc and Shadoo. Such an experiment is also independent of fractionation.

  1. Finally, the suggestion that calnexin-assisted folding of PrP and Shadoo would dictate whether the proteins are found in rafts or not is not supported by any evidence in the manuscript. This is an interesting idea, but as it stands I would advise caution with proposing such a mechanism. I therefore ask the authors to rephrase the last paragraph of their discussion, and perhaps suggest which studies are needed to prove (or disprove) the mechanism that have been proposed. Have the authors considered performing any folding studies for the proteins themselves?

Author Response

Replies to the questions, comments and suggestions raised by Reviewer 2

 We thank Reviewer 2 for reviewing our manuscript and for providing valuable comments and suggestions, which we believe, significantly contributed to the value of the manuscript. We specifically thank the prompt reviewing and for the short time within which we received the comments.

Our point-by-point answers to the questions, comments and suggestions raised  are written in blue and are as follows.

Reviewer 2:

Comments and Suggestions for Authors

In their manuscript, Dondapati and colleagues use a detergent-free lipid fractionation method to isolate membrane raft and non-raft fractions, which they use to study the localization of the GPI-anchored prion protein PrP as well as its homolog Shadoo. The authors show that the proteins are present in both raft and non-raft fractions. The data is supported by live cell imaging. Finally, the authors show that the two proteins directly interact with calnexin, an ER-chaperone. The authors postulate that the chaperone could be required for the folding of the two proteins, which would subsequently allow them to localize in and out of lipid rafts.

First and foremost I would like to say that I really appreciate the time and effort the authors have put into carefully explaining their methodology. The manuscript really stands out in that matter. Similarly, the results are very nicely presented and structured. This was a very easy manuscript to read and comprehend on.

 Below are my (minor) comments for the work:

  1. The English language needs revising, especially in the introduction. Please ask a native English speaker to meticulously correct the manuscript, if possible. Attention should be placed especially on the use of articles. The entire manuscript should be carefully proofread and the language corrected where necessary.

Response 1:  We thank the Reviewer for drawing our attention to the presence of language errors in the manuscript. We consider it important to promote the correct usage of English language in the scientific literature, therefore, we opted for requesting the MDPI’s English language correction service in order to have our manuscript corrected. We have implemented these corrections in our revised manuscript. 

  1. In the introduction, the authors introduce PrP on lines 70 – 87. This is a fairly complicated description and I think the readers of the article would appreciate it if the authors included a figure to illustrate the structure and function of PrP and Shadoo. Additionally, this could enable the authors to draw the domain structures of the protein constructs that are relevant for the study.

Response 2:  We agree with the Reviewer that the specified section was hard to follow as written, furthermore, we consider the suggestion of including a helper figure as an excellent idea.

As a response, we rephrased and rearranged the sentences of the description section, to make the content simpler and more logical to follow.

Modifications are at (numbering corresponds to the revised MS, with tracked changes):

lines 72-73, 75-76 and 83-100: rephrased text.

Additionally, we utilized the opportunity and included a helper figure as suggested. Using linear schemes for the amino acid sequences, we depict the important elements of PrPC and Sho on the figure. To indicate the modifications by EGFP/EYFP-tags present in our protein constructs that are important for the study, we decided upon a simple presentation of indicating only the insertion sites for these tags in the proteins’ schemes, while elaborating it in more detail in the legend. We found this being a simple/elegant and also hopefully clear for the readers. 

The modifications are present at:

lines 101-121: new figure (Figure 1) and its corresponding legend as new text.

  1. I ask the authors to be careful with calling the PrP protein “prion”, as there is some evidence that several different proteins can be considered prions. I know that in the field it is common that “prion” is specifically used to refer to PrP, but an effort should be made to call PrP “prion protein” rather than “prion”. Consider changing this nomenclature for the manuscript.

Response 3:  The Reviewer is right in pointing out that the name “prion” is fundamentally associated to the infectious prion particle[1] and recently also to an increasing number of other proteins, such as Aβ peptide, tau, α-synuclein and more recently TDP43[2]  that are being recognized to display prion-like properties and behavior.  Therefore, it is indeed more appropriate to refer to PrP as “prion protein”. We made this change to the nomenclature throughout the revised manuscript, changing “prion” to “prion protein” when referring to the healthy protein.

Modifications are at:

-throughout the manuscript (36 replacements).

  1. Perhaps the authors could indicate in Figure 2 which fractions represent rafts and non-rafts, much like in Figure 5. I think this would be helpful to the readers of the article.

Response 4:  We agree with the Reviewer’s suggestions and accordingly we marked the raft- and non-raft regions on the Western blots of old Figure 2 and replaced the figure with the revised figure in the revised manuscript. Please note, that due to the inclusion of an additional figure, Figure 1, the numbering of the figures has shifted by one compared to the earlier version and the old Figure 2 is now the Figure 3 in the revised manuscript.

Modifications are at:

 Lines 562-564: deletion of old Figure2 and insertion of the revised, now Figure 3 (corresponding to the revised old Figure 2).

  1. The authors refer to Flotillin 1 as a raft marker, but it is clearly present in the non-raft fractions as well. Could the authors maybe explain why this is the case? Is Flotillin 1 really exclusive to rafts?

Response 5:  Flotillins (flotillin-1 and -2) are integral membrane proteins mostly known (reviewed for eg. by Otto and Nichols (2011))[3] as to cluster in rafts at the plasma membrane to regulate clathrin-independent endocytosis pathways, by promoting membrane curvature and the formation of caveolae. They may serve as adaptor molecules in other endocytic pathways too and are reported to participate in a wide variety of cellular processes. Flotillins received their name from being detergent insoluble and typically floating on top of the sucrose gradients. For this reason, they, specifically flotillin-1, is extensively used in the membrane raft fractionation studies to confirm fractions that would belong to raft-type membranes. However, flotillin-1, despite of being enriched in raft fractions, it is not exclusive to rafts but it is present along the gradient, in non-raft fractions as well, although its distribution pattern may slightly vary according to the cell type or organelle type analysed[4–8]. This fact is acknowledged also in some of these studies and it is explained by flotillin-1 being a protein that is present in several locations and cellular compartments[4,9–13]. For this reason, usually also other known non-raft and raft-resident proteins are blotted for, to decide upon the nature of fractions. Here, we followed the raft-criteria of Persuad-Sawin and coworkers[14] to decide from what density the fractions would include mostly raft-type membranes. These authors utilize flotillin-1 (discussing also the fact that it is present in non-rafts as well) in combination with the transferrin receptor protein (typically residing in non-rafts), as a pair to Western blot for, to decide upon the regions enriched mostly in raft- or non-raft type membranes based on their mutual exclusion or presence, i.e., those fractions are considered to contain mostly rafts in which flotillin-1is present, but the transferrin receptor is absent.  We agree with the Reviewer that referring to flotillin-1 as a “marker” of rafts, even though this terminology is widely used in such studies in the literature, it can be misleading, given that here its presence is only necessary but not sufficient. Accordingly, we rephrased the sentences where we referred to flotillin-1 as “marker”.

Modifications are at:

 Lines 411-413; 447-449; 687-689; 702-703; 711;897-898; 1070 (rephrased sentences).

  1. Since cholesterol is only enriched in rafts, rather than specifically found there, did the authors at any point consider testing for the presence of e.g. sphingomyelin or some other lipid that is known to be specific to rafts or at least very highly enriched in rafts?

Response 6:  The Reviewer is right in pointing out that cholesterol is only enriched in the lipid rafts, and there are other lipids, such as sphingolipids, preferentially partitioning to rafts. In our experiments we have not tested for the presence of this type of lipid class in the collected membrane fractions. We focused on testing the criteria for rafts of Persuad-Sawin and coworkers[14], according to which after a density gradient fractionation, those fractions qualify as membrane raft fractions in where the cholesterol amount is high, total protein amount is low, flottilin-1 is present and the transferrin receptor is absent. In this frame, we monitored the amount of cholesterol along the gradient fractions collected. We agree that it could had been indeed more complete and accurate to also include other, raft-specific lipids for testing beside cholesterol. Taking it further, an analysis of the lipid class composition and/or a more in-depth phospholipid headgroup- and molecular species analysis of the actual fractions collected for our samples could had been very interesting to perform, to see the lipid compositional changes along the gradient and the characteristic compositions of the separated fractions. However, extending the analysis in this direction was out of the scope of the present study.

  1. I will not ask the authors to perform further experiments, as the work is already convincing as it stands, but I am strongly asking the authors to consider the following, as it would further support their conclusions: a well-established experiment to look at the localization of proteins in lipid rafts is performed using fluorescence microscopy of giant unilamellar vesicles (GUVs). GUVs present raft formation when the membrane lipid composition is correct, and rafts can be visualized in several ways, such as adding fluorescently labelled sphingomyelin (which localizes to rafts) or using fluorescent cholera toxin (which binds to gangliosides that are in rafts). Thus, fluorescent recombinant proteins that localize to rafts should co-localize with the fluorescent signal of the raft markers. If the authors would perform such a cell-free experiment, it would support their already existing data for the localization of PrPc and Shadoo. Such an experiment is also independent of fractionation.

Response 7:  We thank the Reviewer for the idea and the suggestion to perform model studies applying raft-mimicking giant unilamellar vesicles (GUVs) and reconstituted GUV membrane-protein systems. Such studies would indeed be a great complementation to our in cellulo results. Due to their size comparable to that of cells, GUVs had been successfully used previously to study lateral domain formation of lipids and partitioning characteristics of membrane proteins by using fluorescence microscopy (reviewed in: [15][16]) .  Such studies combined with our fluorescent protein tagged PrP, Sho and calnexin, if optimized, could indeed provide valuable further basis to our findings. One difficulty to such studies, we think, it could be encountered at the extraction of the proteins for their reconstitution into GUVs. Alternately, recombinant proteins could be used and reconstitution of the GPI-anchors through mimetics could be attempted in case of PrP and Sho. Such constructs were made earlier in the literature for PrP[17] and we have also attempted to make such constructs in the past in an attempt to reinsert them into synthetic and cellular membrane (unpublished). We may, consider to perform such experiments as suggested by Reviewer in our future research. 

  1. Finally, the suggestion that calnexin-assisted folding of PrP and Shadoo would dictate whether the proteins are found in rafts or not is not supported by any evidence in the manuscript. This is an interesting idea, but as it stands I would advise caution with proposing such a mechanism. I therefore ask the authors to rephrase the last paragraph of their discussion, and perhaps suggest which studies are needed to prove (or disprove) the mechanism that have been proposed. Have the authors considered performing any folding studies for the proteins themselves?

Response 8:  As suggested by the Reviewer we rephrased the last paragraph of the discussion. We did not intend to propose earlier that calnexin would regulate whether PrP or Sho localizes to rafts or non-rafts, but since in both kind of fractions calnexin is co-immunoprecipitated, we only propose that it is perhaps bound to a population of PrP and Sho that may be unfolded, at least to an extent that requires the presence of this chaperone. We now made this clearer and also we emphasize that this proposition needs further proof. We had not made experiments to test the folding of the proteins. However, an earlier study that we refer by Pepe and coworkers (2017)[18] demonstrated that Sho has a misfolding tendency, exhibiting  proteinase-K resistant aggregates and also binding to the chaperone calreticulin in  normal cellular conditions in some cultured cells. We may infer such a possibility in our case too.

References

  1. Prusiner, S.B. Novel Proteinaceous Infectious Particles Cause Scrapie. Source Sci. New Ser. 1982, 216, 136–144.
  2. Jaunmuktane, Z.; Brandner, S. Invited Review: The role of prion-like mechanisms in neurodegenerative diseases. Neuropathol. Appl. Neurobiol. 2020, 46, 522–545.
  3. Otto, G.P.; Nichols, B.J. The roles of flotillin microdomains - endocytosis and beyond. J. Cell Sci. 2011, 124, 3933–3940.
  4. Kokubo, H.; Helms, J.B.; Ohno-Iwashita, Y.; Shimada, Y.; Horikoshi, Y.; Yamaguchi, H. Ultrastructural localization of flotillin-1 to cholesterol-rich membrane microdomains, rafts, in rat brain tissue. Brain Res. 2003, 965, 83–90.
  5. Dermine, J.F.; Duclos, S.; Garin, J.; St.-Louis, F.; Rea, S.; Parton, R.G.; Desjardins, M. Flotillin-1-enriched Lipid Raft Domains Accumulate on Maturing Phagosomes. J. Biol. Chem. 2001, 276, 18507–18512.
  6. Gkantiragas, I.; Brügger, B.; Stüven, E.; Kaloyanova, D.; Li, X.-Y.; Löhr, K.; Lottspeich, F.; Wieland, F.T.; Helms, J.B. Sphingomyelin-enriched Microdomains at the Golgi Complex. Mol. Biol. Cell 2001, 12, 1819–1833.
  7. Gagescu, R.; Demaurex, N.; Parton, R.G.; Hunziker, W.; Huber, L.A.; Gruenberg, J. The recycling endosome of Madin-Darby canine kidney cells is a mildly acidic compartment rich in raft components. Mol. Biol. Cell 2000, 11, 2775–2791.
  8. Head, B.P.; Patel, H.H.; Insel, P.A. Interaction of membrane/lipid rafts with the cytoskeleton: Impact on signaling and function. Biochim. Biophys. Acta - Biomembr. 2014, 1838, 532–545.
  9. Macdonald, J.L.; Pike, L.J. A simplified method for the preparation of detergent-free lipid rafts. J. Lipid Res. 2005, 46, 1061–1067.
  10. López-Casas, P.P.; Del Mazo, J. Regulation of flotillin-1 in the establishment of NIH-3T3 cell-cell interactions. FEBS Lett. 2003, 555, 223–228.
  11. Reuter, A.; Binkle, U.; Stuermer, C.A.O.; Plattner, H. PrPc and reggies/flotillins are contained in and released via lipid-rich vesicles in Jurkat T cells. Cell. Mol. Life Sci. 2004, 61, 2092–2099.
  12. Smart, E.J.; Ying, Y.S.; Mineo, C.; Anderson, R.G.W. A detergent-free method for purifying caveolae membrane from tissue culture cells. Proc. Natl. Acad. Sci. U. S. A. 1995, 92, 10104–10108.
  13. Song, J.; Ping, L.-Y.; Duong, D.M.; Gao, X.-Y.; He, C.-Y.; Wei, L.; Wu, J.-Z. Native low density lipoprotein promotes lipid raft formation in macrophages. Mol. Med. Rep. 2016, 13, 2087–2093.
  14. Persaud-Sawin, D.A.; Lightcap, S.; Harry, G.J. Isolation of rafts from mouse brain tissue by a detergent-free method. J. Lipid Res. 2009, 50, 759–767.
  15. Litschel, T.; Schwille, P. Protein Reconstitution Inside Giant Unilamellar Vesicles. Annu. Rev. Biophys. 2021, 50, 525–548.
  16. Dimova, R. Giant Vesicles and Their Use in Assays for Assessing Membrane Phase State, Curvature, Mechanics, and Electrical Properties. Annu. Rev. Biophys. 2019, 48, 93–119.
  17. Hicks, M.R.; Gill, A.C.; Bath, I.K.; Rullay, A.K.; Sylvester, I.D.; Crout, D.H.; Pinheiro, T.J.T. Synthesis and structural characterization of a mimetic membrane-anchored prion protein. FEBS J. 2006, 273, 1285–1299.
  18. Pepe, A.; Avolio, R.; Matassa, D.S.; Esposito, F.; Nitsch, L.; Zurzolo, C.; Paladino, S.; Sarnataro, D. Regulation of sub-compartmental targeting and folding properties of the Prion-like protein Shadoo. Sci. Rep. 2017, 7, 1–15.